# Prompt HIV diagnosis and antiretroviral treatment in postpartum women is crucial for prevention of mother to child transmission during breastfeeding: Survey results in a high HIV prevalence community in southern Mozambique after the implementation of Option B+

**Sheila Fernández-Luis**[1,2]*, **Laura Fuente-Soro**[1,2], **Tacilta Nhampossa**[1,3], **Elisa Lopez-Varela**[1,2], **Orvalho Augusto**[1], **Ariel Nhacolo**[1], **Olalla Vazquez**[4], **Anna Saura-Lázaro**[1,2], **Helga Guambe**[5], **Kwalila Tibana**[5], **Bernadette Ngeno**[6], **Adelino José Chingore Juga**[7], **Jessica Greenberg Cowan**[7], **Marilena Urso**[7], **Denise Naniche**[1,2]

**1** Centro de Investigação em Saúde de Manhiça (CISM), Maputo, Mozambique, **2** ISGlobal, Hospital Clínic, Universitat de Barcelona, Barcelona, Spain, **3** Instituto Nacional de Saúde (INS), Maputo, Mozambique, **4** Hospital Clínico Universitario de Santiago de Compostela, Santiago, Spain, **5** Ministério da Saúde de Moçambique (MISAU), Maputo, Mozambique, **6** U.S Centers for Disease Control and Prevention (CDC), Atlanta, Georgia, United States of America, **7** U.S Centers for Disease Control and Prevention (CDC), Maputo, Mozambique

* sheila.fernandez@isglobal.org

## Abstract

### Objective

World Health Organization recommends promoting breastfeeding without restricting its duration among HIV-positive women on lifelong antiretroviral treatment (ART). There is little data on breastfeeding duration and mother to child transmission (MTCT) beyond 24 months. We compared the duration of breastfeeding in HIV-exposed and HIV-unexposed children and we identified factors associated with postpartum-MTCT in a semi-rural population of Mozambique.

### Methods

This cross-sectional assessment was conducted from October-2017 to April-2018. Mothers who had given birth within the previous 48-months in the Manhiça district were randomly selected to be surveyed and to receive an HIV-test along with their children. Postpartum MTCT was defined as children with an initial HIV positive result beyond 6 weeks of life who initiated breastfeeding if they had a first negative PCR result during the first 6 weeks of life or whose mother had an estimated date of infection after the child's birth. Cumulative incidence accounting for right-censoring was used to compare breastfeeding duration in HIV-exposed

**Data Availability Statement:** Data cannot be shared publicly because of ethical restrictions. Data contain potentially sensitive information and national ethics committee (CNBS) does not authorize data sharing without a protocol request specifying the objectives and the researchers who will have access to the data. Data are available under request (contact via llorenc.quinto@isglobal.org) for researchers who meet the criteria for access to confidential data. We are sharing a copy, in both the original language and English of the questionnaire of the study as Supporting Information.

**Funding:** This research has been supported by the President's Emergency Plan for AIDS Relief (PEPFAR) through the Center for Disease Control (CDC) under the terms of CoAg GH000479. The findings and conclusions in this report are those of the author(s) and do not necessarily represent the official position of the CDC. S.F.L. receives a pre-doctoral fellowship from the Secretariat of Universities and Research, Ministry of Enterprise and Knowledge of the Government of Catalonia and cofounded by European Social Fund. E.L.V. is supported by a Spanish Pediatrics Association (AEP) fellowship and a Ramon Areces Foundation fellowship. For the remaining authors none were declared.

**Competing interests:** The authors have declared that no competing interests exist.

and unexposed children. Fine-Gray regression was used to assess factors associated with postpartum-MTCT.

## Results

Among the 5000 mother-child pairs selected, 69.7% (3486/5000) were located and enrolled. Among those, 27.7% (967/3486) children were HIV-exposed, 62.2% (2169/3486) were HIV-unexposed and for 10.0% (350/3486) HIV-exposure was unknown. Median duration of breastfeeding was 13.0 (95%CI:12.0–14.0) and 20.0 (95%CI:19.0–20.0) months among HIV-exposed and HIV-unexposed children, respectively (p<0.001). Of the 967 HIV-exposed children, 5.3% (51/967) were HIV-positive at the time of the survey. We estimated that 27.5% (14/51) of the MTCT occurred during pregnancy and delivery, 49.0% (2551) postpartum-MTCT and the period of MTCT remained unknown for 23.5% (12/51) of children. In multivariable analysis, mothers' ART initiation after the date of childbirth was associated (aSHR:9.39 [95%CI:1.75–50.31], p = 0.001), however breastfeeding duration was not associated with postpartum-MTCT (aSHR:0.99 [95%CI:0.96–1.03], p = 0.707).

## Conclusion

The risk for postpartum MTCT was nearly tenfold higher in women newly diagnosed and/or initiating ART postpartum. This highlights the importance of sustained HIV screening and prompt ART initiation in postpartum women in Sub-Saharan African countries. Under conditions where HIV-exposed infants born to mothers on ART receive adequate PMTCT, extending breastfeeding duration may be recommended.

## Introduction

Globally in 2019 there were 1.7 million children living with HIV [1]. Children predominantly acquire HIV infection through mother-to-child transmission (MTCT), either during pregnancy, delivery, or breastfeeding. Without preventive interventions, the risk of MTCT is 30–40%, and breastfeeding is responsible for one-third to one-half of these transmissions [2]. However, specific preventive interventions such as antiretroviral treatment (ART) to the mother and antiretroviral prophylaxis to the child can reduce MTCT to less than 5%, even in high HIV burden settings [3, 4].

In order to eliminate MTCT, the Joint United Nations Programme on HIV/AIDS (UNAIDS) and United States President's Emergency Plan for AIDS Relief (PEPFAR) launched the Start Free Stay Free AIDS Free strategy in 2016 [5]. It focuses on the 23 countries with the highest burden of pregnant women and children living with HIV. In 2018, these 23 countries had a rate of MTCT during pregnancy and delivery, (calculated as first 6-week MTCT-rate), of 6.3% [95%CI: 4.9–9.1%] and a rate of MTCT at the end of breastfeeding of 11.8% [95%CI: 9.8–15.2%] [5]. The number of new infections was unevenly distributed; of 130 000 children who acquired HIV in 2018, half came from just six countries (Kenya, Mozambique, Nigeria, South Africa, Uganda and the United Republic of Tanzania) [5].

Breastfeeding contributes substantially to the health, development and survival of young children, particularly in settings with high mortality from diarrhea, pneumonia and malnutrition among children under five years of age [6]. Among HIV-exposed children, breastfeeding has been shown to increase the HIV-free survival at 9 and 18 months of life compared with

formula-feeding [7, 8]. Breastfeeding practices of women living with HIV are influenced by the social circumstances in which mothers make decisions, the fear of transmission, and also the attitudes, knowledge, and updating of health care workers about current guidelines [9–11].

World Health Organization (WHO) adapted its guidelines over time, aiming to balance the benefits of breastfeeding with the risks of HIV transmission. In 2007, WHO recommended exclusive breastfeeding for six months unless replacement feeding was feasible, sustainable and safe for the HIV-positive mothers and their infants [12]. In 2010, the recommendation was extended to 12 months, while antiretrovirals were provided either to the mother or the infant throughout breastfeeding to reduce the risk of postnatal HIV transmission [13]. By 2016, the B + strategy was brought to scale globally, ensuring that the HIV-positive pregnant and breast-feeding women were offered lifelong ART regardless of their CD4 count. At the same time, enhanced post-natal prophylaxis with daily zidovudine (ZDV) and nevirapine (NVP) for a total of 12 weeks was recommended among HIV-exposed children at high risk of acquiring HIV who were breastfeeding [14]. With the expansion of durable ART access for lactating women, WHO again updated its recommendations, advising that HIV+ women should con-tinue to breastfeed for at least 12 months, but ideally for up to 24 months or longer while remaining fully ART adherent [15]. However, this revised guidance promoting extended breastfeeding was based on low to moderate quality evidence [15].

Mozambique adopted the B+ strategy in 2013 [16] and the "test and treat" strategy, in which ART is initiated for all people living with HIV as soon as possible after diagnosis, in 2016 [17]. Nevertheless, in 2018, although more than 95% of HIV-positive pregnant women in Mozambique received ART, the rate of MTCT at the end of breastfeeding was 15.0% [95% CI: 11.8–19.0%]:7% during the first 6-weeks postpartum and 8% during breastfeeding [5], accounting for 10% of global MTCT infections among the Start Free Stay Free AIDS Free pri-ority countries [5]. In 2011, the median duration of total breastfeeding in the general popula-tion of Mozambique was 20.8 months [18]. However, there are no updated data on total duration of breastfeeding disaggregated by mother's HIV-serostatus or on the impact of breastfeeding duration on mother-to-child transmission in Mozambique. Although there are no data about ART adherence and viral load during breastfeeding period at national level, a controlled clinical trial performed in central Mozambique between 2014–2015 among preg-nant and breastfeeding women in B+ showed that without intervention, 52.3%, 46.1% and 38.3% of them returned for 30-day, 60-day and 90-day ART refills, respectively [19]. In the other hand, PEPFAR data showed viral load coverage in pregnant women was approximately 60% and viral suppression 80% at national level in 2020 [20]. Until September 2019, postnatal prophylaxis consisted of 6 weeks NVP to all infants [21, 22] and thereafter enhanced postnatal prophylaxis has been recommended to all HIV exposed infants in Mozambique [23].

We compared the duration of breastfeeding in HIV-exposed and HIV-unexposed children and we identified sociodemographic and HIV-care factors associated with postpartum MTCT, through a cross-sectional household survey in a semi-rural population of southern Mozam-bique with a high HIV community prevalence [24].

## Methods

### Study setting

The study was conducted within the Health and Demographic Surveillance System (HDSS) run by the Manhiça Research Health Center since 1996, which is located in Maputo Province, southern Mozambique [25]. The HDSS platform currently extends over the entire district of Manhiça, which has an area of 2,380 square kilometers and covers 46,441 households and 201,383 inhabitants, each one with a unique identification number. Every household is visited

twice a year to collect data on vital events such as births, deaths, pregnancies and migrations [25]. Verbal autopsies are used to attribute a cause of death to all recorded death events, including those that occurred in the community, in accordance with WHO Verbal Autopsies Instrument Form 2016 [26].

The Manhiça District is served by fifteen health centers, one rural hospital and one referral district hospital. All public health facilities offer free access to HIV care and treatment. Routine patient-level HIV clinical data is recorded by providers in a paper-based system and prospectively entered into an electronic patient tracking system.

At the time of the study, the B+ strategy was already implemented in all the health facilities which provided free ART to HIV-positive pregnant or breastfeeding mothers and 6 week Nevirapine prophylaxis for HIV-exposed children, regardless of both the feeding method and whether the mother's diagnosis and ART initiation occurred during pregnancy or breastfeeding period [21, 27]. The ART regimen that most pregnant and lactating women received during the time period of this study was Tenofovir Disoproxil Fumarate/Lamivudine/Efavirenz (TDF+3TC+EFV) [21, 27].

## Study design and study population

Between October 2017 and April 2018, 5000 of the total children born alive in the previous 48 months within the HDSS were randomly selected to participate in this cross-sectional household survey. After informed consent was obtained, the survey was conducted with mothers, or in case of a mother's absence, migration or death, with the child's primary caregiver. Study HIV counselors administered a specific questionnaire designed to capture sociodemographic characteristics, HIV testing history and ART, antenatal care and duration of breastfeeding.

For each individual mother and child, HIV-status was ascertained through documentation of previous testing, conducting age-appropriate testing with laboratory confirmation or verbal autopsy. Mothers who do not know their status or self-report being HIV-negative were tested at survey, as well as the HIV-exposed children. For children under 18 months of age, HIV diagnosis was determined with molecular testing through HIV DNA Polymerase Chain Reaction (PCR). Children 18 months or older and mothers were tested following the National HIV testing algorithm [21] which included two serial rapid diagnostic tests, Determine [28] and Unigold [29]. Documented known HIV-positive individuals were not re-tested, however for study purposes, all HIV positive participants (including those who were diagnosed prior to or during the study visit) underwent confirmatory testing through Geenius HIV-1/2 Confirmatory Assay [30]. Clinical documentation was also used to obtain information about gestational age and infant antiretroviral prophylaxis. Verbal autopsy from HDSS database was used to ascertain HIV status in children and mothers who had died before the survey. Hospitalizations and outpatients' visits were also obtained through the HDSS database. Information about maternal viral load and CD4 was extracted from the routinely collected data in the electronic patient tracking system, a Microsoft access database [31] co-managed by the Ministry of Health and other stakeholders, where each participant living with HIV had a unique numeric identifier that allows follow-up through the continuum of care [32].

## Definitions

HIV exposure was defined as follows: i) a child whose mother had a documented HIV-infection before birth or at the end of breastfeeding (confirmed exposure) and ii) a child born to a self-reported HIV-positive mother for whom the time of the mother's infection could not be determined (probable exposure). Children born from HIV negative mothers were considered

HIV-unexposed. If the mother was deceased and her HIV-status could not be confirmed, the child´s exposure was considered unknown and were excluded from the analysis.

Date of HIV infection in the mother was estimated as follows:

1. In case of documentation of a previous HIV-negative test, date of infection was assumed to be equal to the midpoint between the last negative HIV test and the day of the survey for mothers of children who are less than 23 months of age. If the interval between the test and the survey was less than 24 months, the midpoint between last negative HIV test and first positive test was used. If the interval was greater than 24 months, the date of seroconversion was not estimated due to the larger uncertainty in the estimation.

2. In case of documentation of a previous HIV-positive test, this was assumed to be the date of infection.

3. In case of no previous documentation and a first positive test on the day of the survey, time of infection was defined as the midpoint between serosurvey and date of most recent delivery.

MTCT was assumed to occur during pregnancy and delivery if the child had a positive PCR result during the first 6 weeks of life [15, 33, 34]. Postpartum MTCT was defined for children with an initial HIV positive result beyond 6 weeks of life who initiated breastfeeding if 1) they had a first negative PCR during the first 6 weeks of life, or 2) did not have a prior negative PCR but whose mother had an estimated date of infection after the child's birth. For children born to mothers with date of infection prior to child's birth but without a DNA PCR by 6 weeks of age, the date of MTCT was considered unknown.

Breastfeeding included any type of breastfeeding (exclusive, mixed and any breastfeeding after the introduction of complementary feeding) since birth. The mother or caregiver self-reported the total duration of any breastfeeding in months at the time of survey.

## Statistical analysis

Medians and interquartile ranges (IQR) were calculated to describe continuous variables and categorical variables were summarized using frequencies and its 95% confidence intervals. Comparisons between groups were made using Pearson chi-square or Fisher exact test and Kruskal Wallis tests, as applicable. In addition, we performed two analyses:

First, we estimated breastfeeding duration in HIV-exposed and HIV-unexposed children with cumulative incidence of breastfeeding cessation, accounting for right censoring. Children who had not initiated breastfeeding were excluded from this analysis. HIV-exposure was evaluated as a factor associated with breastfeeding duration through Fine-Gray regression, using mortality as competing risk and adjusted for age and sex in a multivariable model.

Second, among HIV-exposed children who had been breastfed at any time, we performed a Fine-Gray regression analysis to assess factors associated with postpartum MTCT, adjusting for age and sex and considering mortality a competing risk factor. Infants with MTCT during pregnancy and delivery and children in which it was not possible to establish whether MTCT was during pregnancy and delivery or postpartum were excluded from this analysis. A multivariable model was built including the variables with a p-value lower than 0.20 in the bivariate analysis and with less than 20% missing values. Time-varying covariates were handled by episode splitting. Variables age of the child, sex of the child, mother ART initiation and breastfeeding duration were forced-in covariates due to their clinical relevance. The variable 'mother ART initiation' was treated as a binary variable: ART initiation before delivery yes/no, and had more than 20% missing values. The missing data was addressed through multiple imputation

using a logistic regression imputation method including our outcome variable and the other predictor variables. A total of 20 imputations were performed.

Data was analyzed using Stata statistical software version 16 (Stata Corp., College Station, Texas, USA) [35].

We conducted a sensitivity analysis considering the time of infection of the mother as random date selected from a uniform distribution, a point at the quarter of the interval between the two dates considered at definition and a point at the three-quarters of the interval between the two dates specified above in definitions section.

We conduct another two sensitivity analysis considering the 61 children HIV-exposed with unknown HIV serostatus as HIV-positive and the 12 children HIV-positive with no information on time of HIV acquisition as postpartum MTCT, respectively.

### Ethics statement

This study was approved by the Mozambican National Bioethics Committee and the Barcelona Hospital Clinic Institutional Review Board. It was also reviewed in accordance with CDC human research protection procedures and was determined to be research, but CDC investigators did not interact with human subjects or have access to identifiable data or specimens for research purposes. Written informed consent was obtained from the mothers/caregivers of all children for the mothers/caregiver and children participation. In case of mothers between 14–16 years old, informed consent was provided by the legal representative of the young mother, after the mother's consent.

## Results

Among the 5000 mother/caregiver-child pairs randomly selected for participation, 4826 children were eligible (96.5%) and 174 children were older than 54 months at the time of the visit, and excluded of the study. A total of 1340 mother/child pairs were not located and 3486 were enrolled. Among those, 27.7% (967/3486) children were considered to be HIV-exposed (probable or confirmed), 62.2% (2169/3486) were HIV-unexposed and for 10.0% (350/3486) HIV-exposure was unknown (Fig 1). Taking into account the estimated date of infection of the mother according to assumptions in methods, 77.7% (751/967) were considered exposed to HIV during pregnancy and delivery, 13.2% (128/967) were exposed only in the postpartum period and 9.1% (88/967) were unknown. Among the children exposed only in the postpartum period, 47.6% (61/128) were characterized as HIV-exposed based on maternal HIV testing performed during the study visit and 14% (18/128) were still breastfeeding when their mothers started on ART.

Baseline characteristics differed between HIV-positive and HIV-negative mothers, and their respective children, with the exception of place of birth, child's gender, gestational age and age of the child at the time of study visit (Table 1).

HIV positive mothers were significantly more absent from the household at the time of survey with the caregiver responding, as compared to HIV-negative mothers (8.8% vs 0.0%, p<0.001). HIV-positive mothers were significantly older, with a median age of 28.7 years (IQR: 23.4–33.4) compared with HIV-negative mothers whose median age at survey was 22.6 years (IQR: 18.8–29.3), p<0.001. Almost all the mothers had attended at least one antenatal visit, but the proportion was higher among the HIV-positive mothers (98.1% vs 93.0%, p<0.001).

Only 34.1% (330/967) of HIV-positive mothers had at least one viral load result at the time of survey. However, 75.8% (250/330) of the mothers with viral load results were virally suppressed.

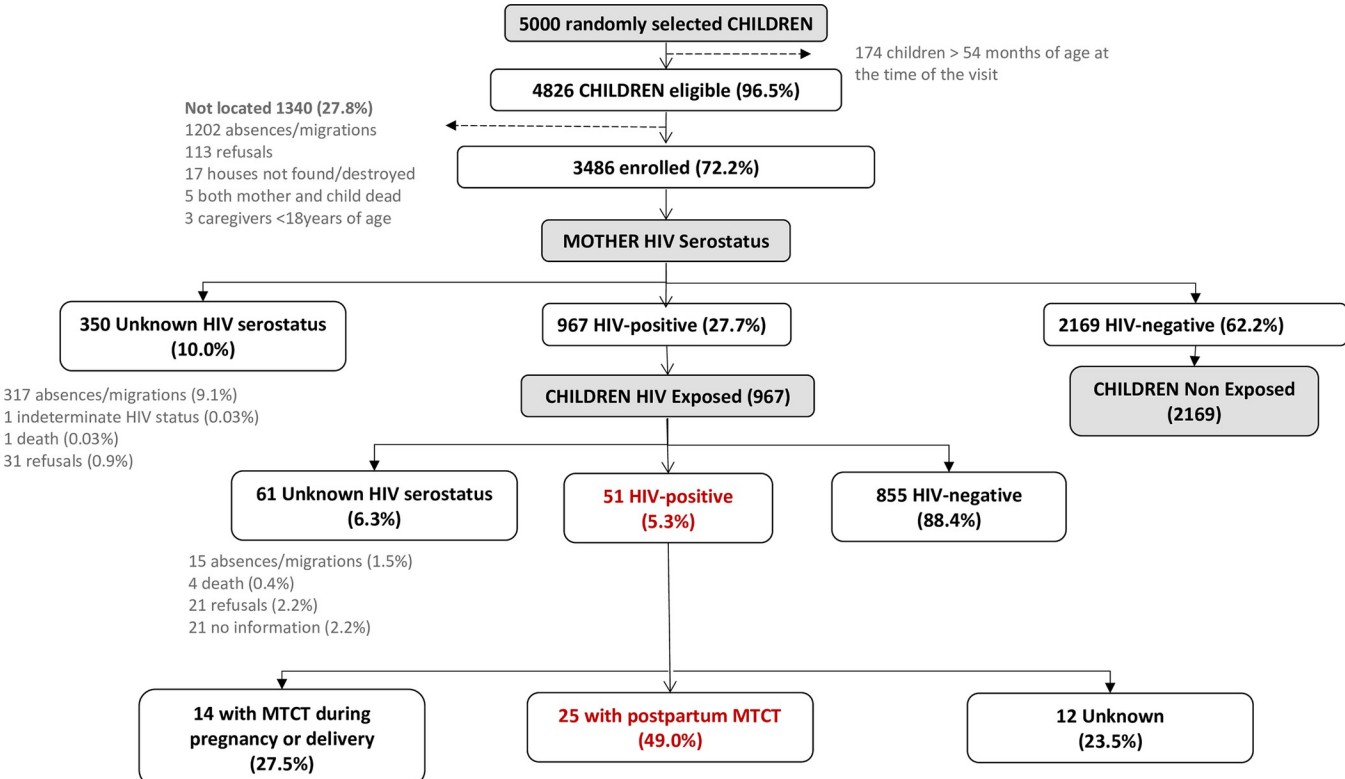

**Fig 1. Study profile among the 5000 mother-child pairs randomly selected for the study.** Percentages are calculated over the previous parent box. MTCT = maternal to child transmission. MTCT during pregnancy and delivery was defined as having a positive PCR result during the first 6 weeks of life. Postpartum MTCT was defined for children with an HIV positive result beyond 6 weeks of life who initiated breastfeeding if 1) they had a first negative PCR during the first 6 weeks of life, or 2) did not have a prior negative PCR but whose mother had an estimated date of infection after the child's birth.

## Breastfeeding duration among HIV-exposed and HIV-unexposed children

In our cohort, the proportion of children who had never breastfed was significantly higher among HIV-exposed children (2.9%, 28/967), than among HIV-unexposed children (0.5%, 11/2169), p<0.001 (Table 1).

The median duration of breastfeeding was 13.0 (95%CI: 12.0–14.0) and 20.0 months (95% CI: 19.0–20.0) among HIV-exposed and HIV-unexposed children respectively, and the risk of discontinuing breastfeeding was almost two-fold higher among HIV-exposed children [adjusted Sub-Hazard Ratio (aSHR) 1.85 (95%CI: 1.67–2.05), p<0.001] (Fig 2). The cumulative incidence of HIV-exposed children breastfeeding dropped at 6 months as compared to unexposed children. At 12 months, only 56.4% (95%CI: 53.2%-59.5%) of HIV-exposed infants were breastfeeding as compared to 77.3% (75.4%-79.0%) of unexposed infants. The gap in breastfeeding between HIV exposed and unexposed children continued through to 18 months of age.

## Postpartum MTCT and associated factors

Of the 967 HIV-exposed children, 5.3% (51/967) were HIV-positive, 88.4% (855/967) were HIV-negative and 6.2% (61/967) with unknown serostatus at the time of the survey. Among the HIV-positive, according to the definitions in methods, we estimated that 49.0% (25/51) of the MTCT occurred postpartum, 27.5% (14/51) during pregnancy and delivery and for 23.5% (12/51) of children the period of infection remained unknown.

**Table 1. Characteristics of HIV-exposed and HIV-unexposed children at the time of the survey (n = 3136).** The 350 children for whom HIV exposure could not be ascertained are not included.

| | | HIV-UNEXPOSED (N = 2169) | | HIV-EXPOSED (N = 967) | | TOTAL (N = 3136) | | |
|---|---|---|---|---|---|---|---|---|
| | | N | % | N | % | N | % | p value |
| MOTHER | | | | | | | | |
| **Age of the mother at delivery in years (IQR)**[*] | | 22.6 (18.8–29.3) | | 28.7 (23.4–33.4) | | 24.8 (19.7–31.2) | | <0.001 |
| **Mother located during the household survey**[**] | Yes | 2164 | 99.8% | 871 | 90.1% | 3035 | 96.8% | <0.001 |
| | Absent or migrated | 0 | 0.0% | 85 | 8.8% | 85 | 2.7% | |
| | Died | 5 | 0.2% | 11 | 1.1% | 16 | 0.5% | |
| **Education level**[**] | Illiteracy | 295 | 13.6% | 189 | 19.5% | 484 | 15.4% | <0.001 |
| | Primary | 1555 | 71.7% | 674 | 69.7% | 2229 | 71.1% | |
| | Seconday or higher | 317 | 14.6% | 91 | 9.4% | 408 | 13.0% | |
| | Unknown | 2 | 0.1% | 13 | 1.3% | 15 | 0.5% | |
| **Marital status**[***] | Single | 265 | 12.2% | 121 | 12.5% | 386 | 12.3% | <0.001 |
| | Married | 1737 | 80.1% | 693 | 71.7% | 2430 | 77.5% | |
| | Divorced/Widowed | 167 | 7.7% | 153 | 15.8% | 320 | 10.2% | |
| **Main source of Income**[**] | Domestic | 66 | 3.0% | 10 | 1.0% | 76 | 2.4% | <0.001 |
| | No fix salary/agriculture | 974 | 44.9% | 497 | 51.4% | 1471 | 46.9% | |
| | Fix Salary | 1129 | 52.1% | 458 | 47.4% | 1587 | 50.6% | |
| | Unknown | 0 | 0.0% | 2 | 0.2% | 2 | 0.1% | |
| **Parity**[**] | Primipara | 723 | 33.3% | 136 | 14.1% | 859 | 27.4% | <0.001 |
| | Secundipara | 458 | 21.1% | 184 | 19.0% | 642 | 20.5% | |
| | Multipara | 987 | 45.5% | 647 | 66.9% | 1634 | 52.1% | |
| | Unknown | 1 | 0.0% | 0 | 0.0% | 1 | 0.0% | |
| **Antenatal clinic visit**[**] | No | 152 | 7.0% | 18 | 1.9% | 170 | 5.4% | <0.001 |
| | Yes | 2017 | 93.0% | 949 | 98.1% | 2966 | 94.6% | |
| CHILD | | | | | | | | |
| **Child found** | Yes | 2107 | 97.1% | 920 | 95.1% | 3027 | 96.5% | **0.018** |
| | Absent or migrated | 23 | 1.1% | 15 | 1.6% | 38 | 1.2% | |
| | Died | 39 | 1.8% | 32 | 3.3% | 71 | 2.3% | |
| **Age at survey in months (IQR)** | | 23.5 (14.5–35.3) | | 24.6 (15.7–36.4) | | 23.9 (14.9–35.7) | | 0.053 |
| **Gender** | Female | 1126 | 51.9% | 479 | 49.5% | 1605 | 51.2% | 0.218 |
| | Male | 1043 | 48.1% | 488 | 50.5% | 1531 | 48.8% | |
| **Born in Mozambique** | No | 38 | 1.8% | 24 | 2.5% | 62 | 2.0% | 0.175 |
| | Yes | 2131 | 98.2% | 943 | 97.5% | 3074 | 98.0% | |
| **Gestational Age** | <37 weeks | 153 | 7.1% | 79 | 8.2% | 232 | 7.4% | 0.088 |
| | ≥ 37 weeks | 1243 | 57.3% | 514 | 53.2% | 1757 | 56.0% | |
| | Unknown | 773 | 35.6% | 374 | 38.7% | 1147 | 36.6% | |
| **Birth order of baby** | 1–3 | 1524 | 70.3% | 538 | 55.6% | 2062 | 65.8% | <0.001 |
| | >3 | 644 | 29.7% | 429 | 44.4% | 1073 | 34.2% | |
| | Unknown | 1 | 0.0% | 0 | 0.0% | 1 | 0.0% | |
| **Breastfeeding at any time** | No | 11 | 0.5% | 28 | 2,9% | 39 | 1.2% | <0.001 |
| | Yes | 2158 | 99.5% | 939 | 97,1% | 3097 | 98.8% | |

IQR: Interquartile Range.

[*]Kruskal Wallis test.

[**]Fisher exact test.

[***]Pearson chi-square

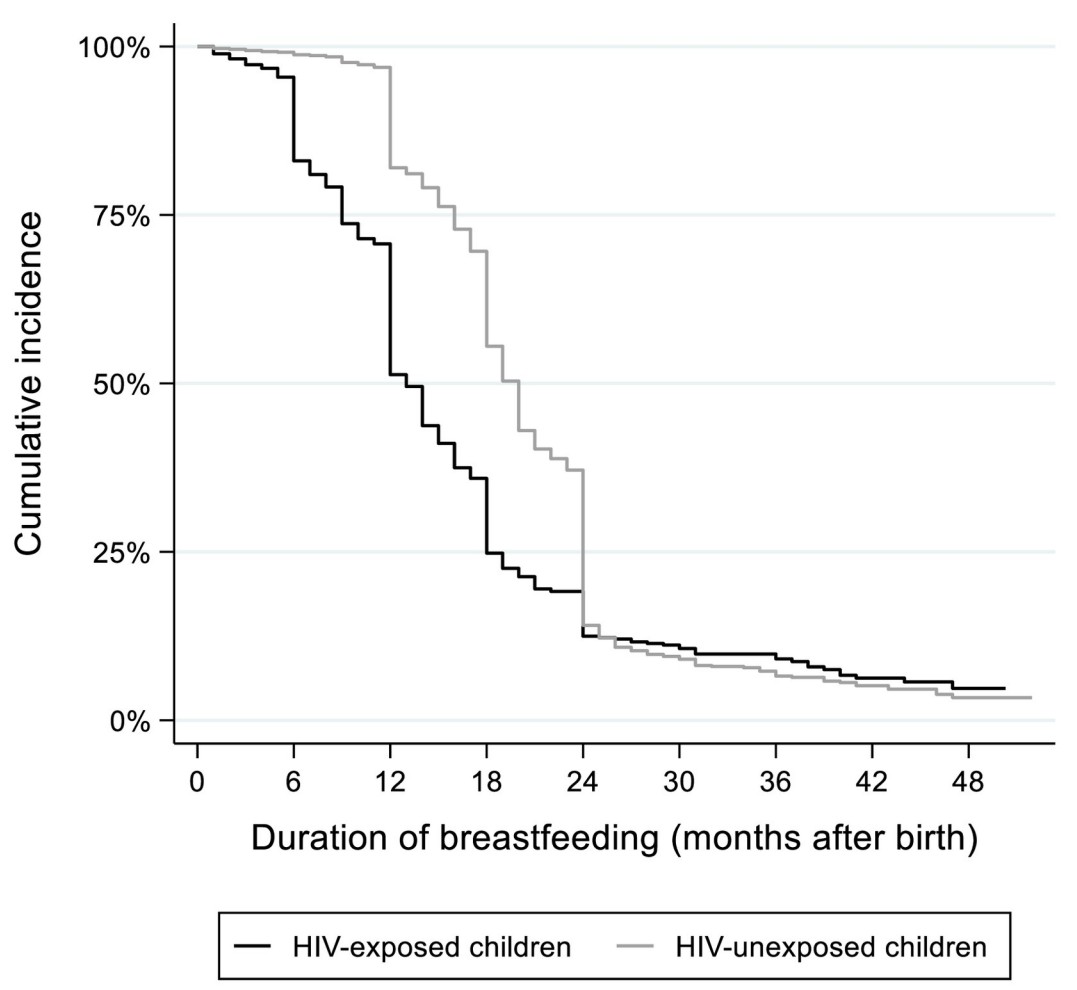

| | Duration of breastfeeding (months after birth) | | | | | | | | |
|---|---|---|---|---|---|---|---|---|---|
| | | 0 | 6 | 12 | 18 | 24 | 30 | 36 | 42 | 48 |
| Number of Children at risk | HIV_unexposed | 2158 | 2035 | 1672 | 929 | 353 | 66 | 40 | 22 | 6 |
| | HIV-exposed | 939 | 861 | 539 | 220 | 95 | 43 | 27 | 14 | 4 |
| | TOTAL | 3097 | 2896 | 2211 | 1149 | 448 | 109 | 67 | 36 | 10 |

**Fig 2. Breastfeeding duration among HIV-exposed and HIV-unexposed breastfed children.** Cumulative incidence was expressed as the proportion of children breastfeeding in a given time period after birth accounting for right censoring and adjusted by sex and age of the child. Children who had not initiated breastfeeding were excluded from this analysis. N = 3097.

Among the 61 children with unknown HIV serostatus, 62.3% (38/61) were female, 4 never breastfed and among the 57 who initiated breastfeeding, the median duration of breastfeeding was 12 (95%CI: 10.4–15.1) months. From 12 of children with unknown period of MTCT, 41.7% (5/12) were female, and all had initiated breastfeeding with the median duration being 15.0 months, (95%CI: 5.6–24.0).

The estimation of the period of MTCT among the 51 HIV-positive children remained the same regardless of the method of estimating the infection time of the mother (as random date selected from a uniform distribution, as a point at the quarter of the interval between the two dates considered at definition and as a point at the three-quarters of the interval between the two dates specified above in definitions section).

Table 2 shows the sociodemographic and clinical characteristics among children with postpartum MTCT compared to HIV-exposed uninfected children, after the exclusion of children with unknown HIV-status, children with MTCT during pregnancy or delivery, children with HIV infection without attribution to pregnancy, delivery or breastfeeding and children who never initiated breastfeeding.

Mothers with postpartum MTCT were mostly diagnosed (60.0%) and initiated ART (52.0%) after the child was born, compared with the 12.3% and 11.4% of diagnosis and ART initiation after the birth among the mothers of HIV exposed uninfected children, p<0.001. Furthermore, only 52.0% (13/25) of children with postpartum MTCT had received antiretroviral prophylaxis at any time after birth compared to 82.8% (688/831) of HIV-exposed uninfected children, p<0.001.

A total of 80.0% (20/25) of children with postpartum-MTCT had attended at least one unscheduled outpatient visits before the survey and a total of 28.0% (7/25) had been hospitalized, compared with the 55.0% (457/831) and 5.8% (48/831) of the HIV exposed uninfected children p = 0.021 and p = 0.001, respectively.

Table 3 shows bivariable and multivariable analysis of factors associated with postpartum MTCT after adjusting for child's sex and age at survey. Children with unknown HIV-status, children with MTCT during pregnancy or delivery and children who never initiated breastfeeding were excluded from this analysis.

Our results show that the duration of breastfeeding was not associated with postpartum MTCT (aSHR: 0.99 [95%CI: 0.96–1.03], p = 0.707). By contrast, children born from mothers who initiated ART at any time after delivery were more likely to acquire HIV postpartum (aSHR: 9.39 [95%CI: 1.75–50.31], p = 0.009).

The estimations obtained in postpartum MTCT related factors were not impacted when we performed sensitivity analysis including the children with unknown HIV status (61 children) or the children with unknown period of HIV infection (12 children): breastfeeding was not associated with postpartum MTCT (aSHR: 0.99 [95%CI: 0.96–1.02] p = 0.547 and aSHR: 0.99 [95%CI: 0.96–1.03] p = 0.660, respectively) and children born from mothers who initiated ART at any time after delivery were more likely to acquire HIV postpartum (aSHR: 8.77 [95% CI: 1.61–47.77] p = 0.012 and aSHR: 8.67 [95%CI: 1.64–45.87] p = 0.011, respectively).

## Discussion

In this study, HIV-exposed children breastfed for significantly less time and had a nearly two-fold higher risk of discontinuation of breastfeeding over 48 months compared with non-exposed children. Evidence of association between breastfeeding duration and postpartum MTCT was not observed. In contrast, mother ART initiation after the date of child birth was associated with a nearly ten-fold higher risk of postpartum MTCT (aSHR: 9.39 [95%CI: 1.75–50.31], p = 0.009).

**Table 2. Mother and child clinical and sociodemographic characteristics of HIV-exposed uninfected children and HIV-exposed children who acquired HIV through postpartum MTCT.** N = 856. Children who never breastfed, children with HIV unknown status, children who acquired HIV through MTCT during pregnancy and delivery and HIV positive children with no information on time of HIV acquisition were excluded from this analysis.

| Characteristics | | HIV-exposed uninfected children (N = 831) | | HIV exposed children with postpartum MTCT (N = 25) | | TOTAL (N = 856) | | p value |
|---|---|---|---|---|---|---|---|---|
| | | N | % | N | % | N | % | |
| MOTHER | | | | | | | | |
| Age of the mother at delivery in years. Median (IQR)* | | 28.7 (23.4–33.4) | | 26.5 (22.4–33.0) | | 28.7 (23.4–33.3) | | 0.577 |
| Mother located during the household survey** | yes | 748 | 90.0% | 23 | 92.0% | 771 | 90.1% | 1.000 |
| | absent or migrated | 75 | 9.0% | 2 | 8.0% | 77 | 9.0% | |
| | died | 8 | 1.0% | 0 | 0.0% | 8 | 0.9% | |
| Education level** | Illiteracy | 171 | 20.6% | 3 | 12.0% | 174 | 20.3% | 0.403 |
| | Primary | 579 | 69.7% | 18 | 72.0% | 597 | 69.7% | |
| | Seconday or higher | 68 | 8.2% | 4 | 16.0% | 72 | 8.4% | |
| | Unknown | 13 | 1.6% | 0 | 0.0% | 13 | 1.5% | |
| Marital status** | Single | 112 | 13.5% | 1 | 4.0% | 113 | 13.2% | 0.321 |
| | Married | 586 | 70.5% | 21 | 84.0% | 607 | 70.9% | |
| | Divorced/Widowed | 133 | 16.0% | 3 | 12.0% | 136 | 15.9% | |
| Main source of Income** | Domestic | 6 | 0.7% | 0 | 0.0% | 6 | 0.7% | 0.456 |
| | No fix salary/agriculture | 428 | 51.5% | 16 | 64.0% | 444 | 51.9% | |
| | Fix Salary | 395 | 47.5% | 9 | 36.0% | 404 | 47.3% | |
| | Unknown | 2 | 0.2% | 0 | 0.0% | 2 | 0.2% | |
| Parity** | Primipara | 116 | 14.0% | 4 | 16.0% | 120 | 14.0% | 0.906 |
| | Secundipara | 152 | 18.3% | 4 | 16.0% | 156 | 18.2% | |
| | Multipara | 563 | 67.7% | 17 | 68.0% | 580 | 67.8% | |
| Antenatal clinic visit** | No | 14 | 1.7% | 1 | 4.0% | 15 | 1.8% | 0.361 |
| | Yes | 817 | 98.3% | 24 | 96.0% | 841 | 98.2% | |
| Mother HIV diagnosis** | Before the date of childbirth | 653 | 78.6% | 9 | 36.0% | 662 | 77.3% | <0.001 |
| | After the date of childbirth | 102 | 12.3% | 15 | 60.0% | 117 | 13.7% | |
| | Unknown | 76 | 9.1% | 1 | 4.0% | 77 | 9.0% | |
| Mother ART initiation*** | Before the date of childbirth | 513 | 61.7% | 6 | 24.0% | 519 | 60.6% | <0.001 |
| | After the date of childbirth | 95 | 11.4% | 13 | 52.0% | 108 | 12.6% | |
| | Unknown | 223 | 26.8% | 6 | 24.0% | 229 | 26.8% | |
| Mother CD4 at childbirth** | <200 cel/mm3 | 42 | 5.1% | 4 | 16.0% | 46 | 5.4% | 0.083 |
| | 200–500 cel/mm3 | 175 | 21.1% | 5 | 20.0% | 180 | 21.0% | |
| | >500 cel/mm3 | 327 | 39.4% | 6 | 24.0% | 333 | 38.9% | |
| | Unknown | 287 | 34.5% | 10 | 40.0% | 297 | 34.7% | |
| Mother viral load at childbirth** | <1000copies/ml | 247 | 29.7% | 7 | 28.0% | 254 | 29.7% | 0.352 |
| | ≥1000copies/ml | 45 | 5.4% | 3 | 12.0% | 48 | 5.6% | |
| | Unknown | 539 | 64.9% | 15 | 60.0% | 554 | 64.7% | |
| CHILD | | | | | | | | |
| Child located during the household survey** | yes | 813 | 97.8% | 20 | 80.0% | 833 | 97.3% | <0.001 |
| | absent or migrated | 2 | 0.2% | 0 | 0.0% | 2 | 0.2% | |
| | died | 16 | 1.9% | 5 | 20.0% | 21 | 2.5% | |
| Age at survey in months. Median (IQR) | | 24.6 (15.6–35.6) | | 30.0 (20.9–34.2) | | 24.6 (15.7–36.5) | | 0.369 |
| Gender*** | Female | 420 | 50.5% | 9 | 36.0% | 429 | 50.1% | 0.152 |
| | Male | 411 | 49.5% | 16 | 64.0% | 427 | 49.9% | |

(*Continued*)

**Table 2.** (Continued)

| Characteristics | | HIV-exposed uninfected children (N = 831) | | HIV exposed children with postpartum MTCT (N = 25) | | TOTAL (N = 856) | | p value |
|---|---|---|---|---|---|---|---|---|
| | | N | % | N | % | N | % | |
| **Born in Mozambique**** | No | 19 | 2.3% | 1 | 4.0% | 20 | 2.3% | 0.451 |
| | Yes | 812 | 97.7% | 24 | 96.0% | 836 | 97.7% | |
| **Gestational Age**** | <37 weeks | 67 | 8.1% | 2 | 8.0% | 69 | 8.1% | 0.345 |
| | ≥ 37 weeks | 451 | 54.3% | 10 | 40.0% | 461 | 53.9% | |
| | Unknown | 313 | 37.7% | 13 | 52.0% | 326 | 38.1% | |
| **Birth order of baby** *** | 1–3 | 463 | 55.7% | 13 | 52.0% | 476 | 55.6% | 0.713 |
| | >3 | 368 | 44.3% | 12 | 48.0% | 380 | 44.4% | |
| **Number of Outpatient visits before survey**** | 0 | 374 | 45.0% | 5 | 20.0% | 379 | 44.3% | **0.021** |
| | 1 | 92 | 11.1% | 6 | 24.0% | 98 | 11.5% | |
| | ≥2 | 365 | 43.9% | 14 | 56.0% | 379 | 44.3% | |
| **Number of Hospitalizations before survey**** | 0 | 783 | 94.2% | 18 | 72.0% | 801 | 93.6% | **0.001** |
| | 1 | 39 | 4.7% | 5 | 20.0% | 44 | 5.1% | |
| | ≥2 | 9 | 1.1% | 2 | 8.0% | 11 | 1.3% | |
| **Duration of breastfeeding in months. Median (IQR)** | | 12.0 (8.0–17.0) | | 12.2 (12.0–18.0) | | 12.0 (8.0–17.0) | | 0.310 |
| **Received Antiretroviral prophylaxis**** | Yes | 688 | 82.8% | 13 | 52.0% | 132 | 15.4% | **<0.001** |
| | No | 20 | 2.4% | 3 | 12.0% | 701 | 81.9% | |
| | Unknown | 123 | 14.8% | 9 | 36.0% | 23 | 2.7% | |

IQR: Interquartile Range.

*Kruskal Wallis test.

**Fisher exact test.

***Pearson chi-square

Sociodemographic data (Age, Eductional level, Marital status, Income, Parity), antenatal clinic visits and breastfeeding information were self-reported.

HIV data were obtained through medical documentation (Mother HIV diagnosis, Mother ART initiation) or HIV database (Mother cd4 at child birth, Mother viral load at child birth). Hospitalizations and outpatients' visits were obtained through HDSS database.

Gestational Age and Infant Antiretroviral prophylaxis were obtained through medical documentation.

Our results demonstrate that children born from mothers who initiated ART after childbirth had a higher risk of acquiring HIV during the breastfeeding period, as previously indicated by other studies [36]. At the time of the study, the B+ strategy was already implemented and lifelong ART was recommended to HIV-positive pregnant or breastfeeding mothers [21], [27]. However, success of the B+ strategy in reaching all pregnant and breastfeeding women is highly dependent on a sustained frequency of HIV testing not only during pregnancy but during the post-partum period. The reasons for not initiating ART could be multiple: others may not have initiated lack of awareness of HIV status, death, not willing or because of service delivery shortfalls or stockouts. A study conducted in southern Mozambique between 2008 to 2011 before B+ implementation, found an HIV incidence in women of 3.2/100 women-years (95%CI: 2.30–4.46) in breastfeeding women during the postpartum period. In absence of treatment, this was reflected by a postpartum-MTCT rate of 21% at 18-months of age among their children [37]. In Mozambique, at the time of the study, re-testing in all pregnant women every three months during pregnancy was recommended, however, delivery and the postpartum period were not targeted time points for re-testing [38, 39]. Our results suggest that establishing specific retesting times during the postpartum period in areas of high HIV incidence could

**Table 3. Mother and child clinical and sociodemographic risk factors associated with postpartum MTCT among HIV-exposed children.** N = 856. Fine-Gray subdistribution hazard regression with death as a competing risk was conducted. The same exclusion factors as those described in Table 2 were applied. The multivariable model was built of including the variables with a p-value lower than 0.20 in the bivariate analysis and with less than 20% missing values. Variables age of child, mother ART initiation and breastfeeding duration were forced-in covariates due to their clinical relevance. Multiple imputation was performed in mother ART initiation. Mother HIV diagnosis was excluded because of collinearity.

| Factors | | Univariable Model* | | | Multivariable Model** | | |
|---|---|---|---|---|---|---|---|
| | | SHR | (95% Conf. Interval) | p-value | aSHR | (95% Conf. Interval) | p-value |
| **Mother** | | | | | | | |
| Age of the mother at delivery (in years) | | 0.99 | 0.93–1.05 | 0.627 | | | |
| Education level (N = 843) | No education | 1 | | 0.284 | | | |
| | Basic | 1.76 | 0.52–5.96 | | | | |
| | Medium/High | 3.31 | 0.74–14.82 | | | | |
| Marital status | Single | 1 | | 0.333 | | | |
| | Married | 3.95 | 0.53–29.39 | | | | |
| | Divorced/Widowed | 2.52 | 0.26–24.24 | | | | |
| Parity | Primipara | 1 | | 0.931 | | | |
| | Secundipara | 0.77 | 0.19–3.07 | | | | |
| | Multipara | 0.87 | 0.29–2.60 | | | | |
| Antenatal clinic visit | No | 1 | | 0.398 | | | |
| | Yes | 0.44 | 0.06–3.01 | | | | |
| Mother HIV diagnosis[1] (N = 779) | before the date of childbirth | 1 | | **0.008** | | | |
| | after the date of childbirth | 4.20 | 1.46–12.06 | | | | |
| Mother ART initiation[1] | before the date of childbirth | 1 | | **<0.001** | | | |
| | after the date of childbirth | 9.18 | 3.87–21.80 | | 9.39 | 1.75–50.31 | **0.009** |
| Mother cd4 at childbirth (N = 559) | <200 cel/mm3 | 1 | | | | | |
| | 200–500 cel/mm3 | 0.32 | 0.09–1.16 | **0.042** | | | |
| | >500 cel/mm3 | 0.20 | 0.06–0.71 | | | | |
| Mother viral load at childbirth (N = 302) | <1000copies/ml | 1 | | | | | |
| | ≥1000copies/ml | 2.30 | 0.60–8.82 | 0.226 | | | |
| **Children** | | | | | | | |
| Age at survey in months | | 1.01 | 0.99–1.04 | 0.348 | 1.01 | 0.97–1.06 | 0.577 |
| Gender | Male | 1 | | 0.160 | 1 | | |
| | Female | 1.80 | 0.79–4.06 | | 1.99 | 0.67–5.96 | 0.218 |
| Gestational Age (N = 530) | <37 weeks | 1 | | 0.699 | | | |
| | ≥ 37 weeks | 0.74 | 0.16–3.40 | | | | |
| Birth order of baby | 1–3 | 1 | | 0.722 | | | |
| | >3 | 1.15 | 0.53–2.52 | | | | |
| Born in Mozambique | No | 1 | | 0.575 | | | |
| | Yes | 0.56 | 0.07–4.24 | | | | |
| Median time of breastfeeding | | 1.01 | 0.98–1.05 | 0.557 | 0.99 | 0.96–1.03 | 0.707 |
| Received Antiretroviral prophylaxis (N = 724) | Yes | 1 | | | 1 | | |
| | No | 7.25 | 2.13–24.73 | **0.002** | 3.01 | 0.55–16.54 | 0.205 |

*N = 856 unless otherwise specified

** N = 818

SHR: subhazard ratio.

aSHR:adjusted subhazard ratio.

1 Time-varying covariates. Time-varying covariates were handled by episode splitting

reinforce the prevention of MTCT in LMIC, particularly in areas of high HIV prevalence where the risk of contracting HIV during the breastfeeding period is high. This would facilitate initiation of ART in breastfeeding mothers and antiretroviral prophylaxis in their HIV exposed infants. In addition to retesting, pre-exposure prophylaxis (PrEP) among women who remain at risk for HIV acquisition in the postpartum period may be an effective approach to reduce MTCT. In 2018 Mozambique began with the pilot implementation of PrEP in sero-discordant couples in Zambezia province and will expand PrEP nationally in 2022 to additional target groups, including pregnant and breastfeeding women at risk and key populations [40].

The vast majority (>97%) of the mothers in our study initiated breastfeeding. This is in agreement with data from Demographic and Health Surveys (2000–2013) for 57 countries which showed a consistently high percentage of children who had ever breastfed across all regions (weighted mean 98.2%, range of countries 87.8–99.8%) [41]. In terms of duration of breastfeeding, in a study conducted in 2019 in an urban population of South Africa, similar to our results, the duration of breastfeeding was also significantly lower among HIV-positive mothers as compared to HIV-negative mothers [3.9 months vs 9.0 months, respectively, p<0.001] [42]. The striking difference in the duration of breastfeeding between women in our study and those studies by Roux et al are likely due to shorter breastfeeding duration in urban compared to rural populations, compounded by social and contextual barriers, as previously described [43, 44]. Advice of health workers, influence of relatives, stigma, conflicting opinions about the risk for MTCT and poor dissemination of policies have been described as main reasons affecting breastfeeding among HIV-positive women in the past [45–47]. However, little is known about the barriers for prolonged breastfeeding after the implementation of the 2016 WHO feeding guidelines.

The higher risk of discontinuing breastfeeding in HIV-exposed children could have important health implications for this population. A clinical trial conducted in Uganda demonstrated higher rates of serious gastroenteritis among HIV-exposed uninfected infants with early breastfeeding cessation (8.0/1000 child-months) when compared to later breastfeeding cessation (3.1/1000 child-months; p<0.001) [48]. In addition, early cessation of breastfeeding is associated with a lower probability of HIV-free survival compared with longer breastfed infants who had lower overall mortality [44].

Our results suggest that in the context of B+ (lifelong ART to the mother and antiretroviral prophylaxis to the children), duration of breastfeeding is not associated with an increased risk of postpartum MTCT. Bispo et al (2017) in their systematic review found a pooled estimated rate of overall HIV transmission by age six months of 3.5% and a pooled postnatal transmission rate by six months of 1.1% in women who were on ART from early-mid pregnancy and breastfed for 6 months [4]. To our knowledge, no such study has been published after the expansion of B+ prevention MTCT programs. Thus, our results fill the gap in knowledge on the risk of MTCT associated with breastfeeding beyond 6 months of age in the context of B + strategy that recommends lifelong antiretroviral treatment for all pregnant and breastfeeding women living with HIV.

Effective strategies to increase the duration of safe breastfeeding in HIV-exposed children could allow them to reap the benefits of breastfeeding through the second year of life such as decreased morbidity and mortality in comparison to HIV exposed infants who are weaned earlier [49, 50]. Highlighted strategies proposed by the literature to promote breastfeeding among HIV-exposed children include increased coverage of extended nevirapine prophylaxis to the infants, active support and breastfeeding counseling [51]. Other strategies such as widespread access to viral load testing could mitigate the fear of HIV transmission, reported by mothers and health workers as a barrier to breastfeeding [52]. In the absence of viral load testing, expansion of antiretroviral prophylaxis among HIV-exposed children until the end of breastfeeding i one strategy supported by literature [44].

This study has several limitations. First, 27% of the randomly selected mother/child pairs were not located and we do not have information about their HIV status. Second, breastfeeding duration was self-reported (in months) by mothers/caregivers for both HIV-exposed and HIV non-exposed children patients at the time of the survey. As we have analyzed children born up to 48 months prior to data collection, potential memory bias could affect the estimations of breastfeeding duration. Third, due to missing data, it was not possible to establish the HIV status of 6.2% (61/967) of HIV exposed children and it was not possible to establish the period of MTCT (during pregnancy and delivery or during breastfeeding) of 23.5% (12/51) of HIV infected children. We conducted a sensitivity analysis considering the 61 children with unknown HIV serostatus as HIV-positive and the 12 HIV-infected children with unknown period of MTCT as postpartum MTCT. It did not impact the estimates of postpartum-MTCT and associated factors, even if the 12 with had a median duration of breastfeeding of 15 months (95%CI: 5.6–24.0) Further studies with larger sample size and breastfeeding periods of 24 months or longer are needed. Fourth, the exact date of mothers' seroconversion was unknown and was established according to the assumptions described in the methods section. A sensitivity analysis considering the time of infection as random date selected from a uniform distribution, a point at the quarter of the interval between the two dates and a point at the three-quarters of the interval between the two dates was robust and did not impact the classification of time of HIV acquisition among HIV-positive children. Fifth, service delivery shortfalls or stockouts of ART were not assessed during the study, which may have affected the ART initiation of among HIV-positive mothers and ARV prophylaxis among HIV-exposed infants. Finally, in Mozambique viral load became routinely available after 2016, and viral load coverage has slowly climbed over time. As a result, more than 60% of HIV infected mothers included in the study did not have any viral load results prior to the survey, thus no adjustments by viral suppression were possible in the multivariable model.

## Conclusion

The risk for postpartum MTCT was nearly tenfold higher in women who were newly diagnosed or initiated ART during the postpartum period as compared to those who initiated ART prior to childbirth. HIV-exposed children breastfed for significantly less time compared with non-exposed children. Breastfeeding duration was not observed to be associated with an increased risk of MTCT in the postpartum period in women on ART. These results emphasize the importance of repeat HIV testing in the postpartum period, and support PrEP in breastfeeding women at high risk for HIV infection. Moreover, further public health messaging to encourage prolonged breastfeeding among HIV-exposed infants born to mothers adherent to ART in Sub-Saharan African countries could bring long term health benefits for children.

## Supporting information

**S1 Appendix. Study questionnaires in English.**
(ZIP)

**S2 Appendix. Study questionnaires in Portuguese.**
(ZIP)

## Acknowledgments

We acknowledge support from the Spanish Ministry of Science, Innovation and Universities through the "Centro de Excelencia Severo Ochoa 2019–2023" Program (CEX2018-000806-S), and support from the Generalitat de Catalunya through the CERCA Program. We want

specially acknowledge Elisabeth Salvo for their contributions to this work. The authors grate-
fully acknowledge the Ministry of Health of Mozambique, our research team, collaborators,
and especially all communities and participants involved.

## Author Contributions

**Conceptualization:** Elisa Lopez-Varela, Marilena Urso, Denise Naniche.

**Data curation:** Ariel Nhacolo.

**Formal analysis:** Sheila Fernández-Luis, Orvalho Augusto, Olalla Vazquez, Anna Saura-
Lázaro, Denise Naniche.

**Funding acquisition:** Elisa Lopez-Varela, Marilena Urso.

**Investigation:** Tacilta Nhampossa, Olalla Vazquez.

**Methodology:** Laura Fuente-Soro, Tacilta Nhampossa, Denise Naniche.

**Supervision:** Laura Fuente-Soro, Tacilta Nhampossa.

**Writing – original draft:** Sheila Fernández-Luis.

**Writing – review & editing:** Elisa Lopez-Varela, Orvalho Augusto, Helga Guambe, Kwalila
Tibana, Bernadette Ngeno, Adelino José Chingore Juga, Jessica Greenberg Cowan, Mari-
lena Urso, Denise Naniche.

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
