## [Decision Letter · Decision Letter 0]

19 Oct 2021

PONE-D-21-25758Prolonged breastfeeding is safe in HIV-exposed children: Survey results in a high HIV prevalence community in southern Mozambique after the implementation of Option B+PLOS ONE

Dear Dr. Fernandez,

Thank you for submitting your manuscript to PLOS ONE. After careful consideration, we feel that it has merit but does not fully meet PLOS ONE’s publication criteria as it currently stands. Therefore, we invite you to submit a revised version of the manuscript that addresses the points raised during the review process.

The authors have attempted to tackle an important question in the PMTCT world of how much transmission likely occurs at the various stages of the pregnancy/delivery/postpartum periods, and what associations exist with duration of breastfeeding with HIV transmissions. The work, however, as noted by the reviewers, has marked limitations, which dampens my overall enthusiasm.  Duration of breastfeeding is retrospectively self-reported more than 2 years back, if I understand the methods correctly.  Not enough details on the overall sample from which they randomly selected 5000 pairs to follow is not given, nor justification for why 5000.  I am a little confused by their methods and use of proportional hazards modeling when at times events/person time (i.e., Poisson modeling) and not time to event analysis is needed, though the journal may need formal biostatistical review for this point if the authors choose to stay with their current methods.  On a small note, small English formatting and copy editing aspects (e.g. tables split on multiple pages) simply makes it harder to the read this paper with joy.  Ultimately, as one reviewer put it, there is marked over-interpretation of their data.

We look forward to receiving your revised manuscript.

Kind regards,

Rena C Patel, MD, MPH

Academic Editor

PLOS ONE

Journal Requirements:

2. Please include additional information regarding the survey or questionnaire used in the study and ensure that you have provided sufficient details that others could replicate the analyses. For instance, if you developed a questionnaire as part of this study and it is not under a copyright more restrictive than CC-BY, please include a copy, in both the original language and English, as Supporting Information

3. Please include your tables as part of your main manuscript and remove the individual files. Please note that supplementary tables (should remain/ be uploaded) as separate "supporting information" files

Reviewers' comments:

Reviewer's Responses to Questions

**Comments to the Author**

1. Is the manuscript technically sound, and do the data support the conclusions?

Reviewer #1: No

Reviewer #2: Partly

Reviewer #3: Yes

2. Has the statistical analysis been performed appropriately and rigorously? 

Reviewer #1: I Don't Know

Reviewer #2: I Don't Know

Reviewer #3: Yes

3. Have the authors made all data underlying the findings in their manuscript fully available?

Reviewer #1: No

Reviewer #2: Yes

Reviewer #3: Yes

4. Is the manuscript presented in an intelligible fashion and written in standard English?

Reviewer #1: Yes

Reviewer #2: Yes

Reviewer #3: Yes

5. Review Comments to the Author

Reviewer #1: General:

1. This study has the potential to make important contributions to the literature on this subject. With the success of PMTCT programs, vertical transmission rates have been declining; however, MTCT has not been eliminated, and the relative contribution of postpartum/breastfeeding MTCT to the vertical transmission events that continue to occur is not well understood. This study suggests that postpartum/breastfeeding MTCT most commonly occurs in the context of women newly acquiring HIV in the postpartum period and in the absence of infant ARV prophylaxis. The authors emphasize that duration of breastfeeding was not associated with postpartum MTCT, suggesting that prolonged breastfeeding can/should be supported in this context; however, this conclusion does not seem to logically follow from the results. Rather, it seems that the emphasis should be on prevention, diagnosis, and treatment of women who remain at risk for HIV acquisition in the postpartum period. It may be true that prolonged breastfeeding might be safe (assuming that the mother and baby are adherent to ART and ARV prophylaxis, respectively), but these data/findings do not strongly support this conclusion. That all said, the interpretation and messaging need to be modified. If these issues and the following comments are adequately addressed, then I think this article can reconsidered for publication.

Title:

2. I appreciate the sentiment, but to declare prolonged breastfeeding “safe” without further context is too strong. Furthermore, the context was a cross-sectional sample of mother-child pairs, of whom nearly 30% could not be found and therefore had unknown (and perhaps worse) outcomes. Also, among the subset of HIV-exposed infants who were located and tested, there was an overall MTCT rate of 5.3%, more than half of which was attributed to postpartum/breastfeeding. All said, a postpartum/breastfeeding MTCT rate of 2.7% (26/967) should not be celebrated as “safe”. Just because self-reported breastfeeding duration was not associated with post-partum MTCT in your multivariable model, it does not mean that the postpartum MTCT was not attributable to breastfeeding; in fact, it seems that breastfeeding in the absence of maternal HIV diagnosis, maternal ART, and infant ARV prophylaxis might have been the cause for these MTCT events. Consider tempering the title and revising the interpretation and messaging throughout the manuscript.

Abstract:

3. The study design should be more clearly stated. From what I can gather, this is a cross-sectional assessment of an observational cohort.

4. 69.7% (3486/5000) were “found and interviewed”. Were all of these women and their children also tested for HIV, as suggested in the Methods. If so, what was the median age [IQR] of children when they were tested, and were all tested after complete cessation of breastfeeding?

5. How can 10% of children have unknown HIV exposure status when/if all of the mothers were tested for HIV in the context of this study (see Comment #4, above)? Perhaps this is made clear later in the manuscript.

6. “P-value” does not need to be spelled out (e.g., p<0.001 should suffice).

7. For the same reasons outlined in Comment #2 (re: Title), I do not think the Conclusion is well supported by the Results. To me, the results suggest that more needs to be done to promote uptake of infant ARV prophylaxis, and that perhaps when mothers are virally suppressed AND infants are appropriately prophylaxed, then prolonged breastfeeding can and should be supported in this context.

Introduction:

8. In the last sentence of the 5th paragraph (line 52) of the Introduction, there is a typo. Presumably, “2014-1015” should be “2014-2015”.

9. In the 5th paragraph of the Introduction, the authors state that “there is no data about ART adherence and viral load during breastfeeding period at national level…”, so they instead cite Reference #18 to illustrate that there is substantial non-adherence to ART refills in the first 3 months postpartum. Firstly, it should be “there ARE no data…” (plural). Secondly, aren’t there PEPFAR data about retention and viral suppression that can be disaggregated for pregnancy/breastfeeding status (or at least for women of child bearing age) to approximate national statistics?

10. There is sufficient introduction to breastfeeding guidelines for HIV+ women, but there is insufficient background information about HIV-exposed infant ARV prophylaxis guidelines, which have also evolved over time. Since your main finding was that lack of infant ARV prophylaxis was associated with postpartum MTCT, it seems like more background on this needs to be presented (i.e., WHO and Mozambican guidelines).

Methods:

11. In the last paragraph of the Statistical Analysis section of the Methods (line 157), the authors state that there were “61 children with unknown HIV serostatus”, but in the Abstract it is noted that “for 10.1% (350/3486) HIV-exposure was unknown.” This distinction becomes clear once the reader gets to Figure 1, but please clarify this in the text as well.

Results:

12. In the Study Design and Study Population section of the Methods (line 84) it is stated that “5000 children born alive in the previous 48 months were randomly selected”, but in the first sentence of the Results section (line 174) it is stated that 174 children were ineligible/excluded because they were “older than 48 months”. How is this possible?

13. In the second sentence of the Results it is noted that “1340 [27%] mother/child pairs were not located”, but no further explanation is provided in the text. In Figure 1, it is noted that these were mostly “absent/migrant”, but this is not well characterized. Also in Table 1, among those who were included/located, it seems that all of those women who were “absent or migrated” were HIV+ (n=85; 8.8%) vs. HIV- (n=0), but the numbers are different for the HIV-exposed (n=15; 1.6%) vs unexposed infants (n=23; 1.1%). Whatever the case, this might suggest that these were not missing at random. Can the authors provide some additional details as to why mother-child pairs might not have been located, and whether the characteristics of those not located were similar/different to those who ultimately participated in the study? Is it possible that those who were not located were also more likely to be non-adherent to PMTCT care and therefore more likely to have postpartum MTCT? All said, I am concerned that the study findings might not be generalizable based on this limitation. This must be addressed or at least acknowledged as a severe limitation.

14. Very minor point, but in the Results (line 178) it is stated that for 10.0% HIV-exposure was unknown, while in the Abstract it says 10.1%. Please reconcile.

15. In line 181 it is stated that “13.4% (130/967) were exposed only in the postpartum period”. What proportion of these were characterized as such based on maternal HIV testing performed during the study visit? What proportion of these women were diagnosed with HIV and started on ART while the baby was still breastfeeding?

16. In lines 186-188 it is stated “HIV-positive mothers were significantly older, with a median age of 28.7 years (IQR: 23.4-187 33.4) and almost all (98.1%) had attended at least one antenatal visit, when compared with HIV-negative mothers.” Can you please include the statistics for the HIV-negative women for comparison, rather than requiring the reader to cross-reference Table 1.

17. In line 189, it is stated that, “Only 34.1% of HIV-positive mothers had at least one viral load result at the time of survey.” What proportion of these were virologically suppressed?

18. In the Postpartum MTCT and Associated Factors section of the Results (line 208), “6,2%” should be “6.2%”.

19. In the Postpartum MTCT and Associated Factors section of the Results, the sentence “Among the 61 children with unknown HIV serostatus, 62.3% (38/61) were female, 4 never breastfed and among the 57 who initiated breastfeeding, the median duration of breastfeeding was 12 (95%CI: 10.4 - 15.1) months and a total of 53 would meet the definition of postpartum MTCT” (lines 212-215) is quite confusing and needs to be re-written to enhance clarity.

20. In the Postpartum MTCT and Associated Factors section of the Results, the paragraph found in lines 227-232 is essential to contextualizing the findings of this study. The fact that, “Mothers with postpartum MTCT were mostly diagnosed (57.7%) and initiated ART (50.0%) after the child was born, compared with the 12.3% and 11.4% of diagnosis and ART initiation after the birth among the mothers of HIV exposed uninfected children, p-value<0.001” indicates a problem that heretofore has not been very well discussed. This frames the problem not only as postpartum MTCT among mother-child pairs breastfeeding in the context of known HIV status during the antepartum/peripartum period, but rather postpartum HIV diagnosis and ART initiation (and lack thereof) among women who were previously HIV-negative or HIV-unknown status. Similarly, that “only 50.0% (13/26) of children with postpartum MTCT had received antiretroviral prophylaxis at any time after birth compared to 82.8% (688/831) of HIV exposed uninfected children, p-value<0.001” should not come as a surprise when nearly two-thirds of their mothers were not diagnosed with HIV until after the child was born. This further emphasizes that the issue has less to do with prescription and adherence to infant ARV prophylaxis and more to do with mothers not knowing their HIV status and/or not benefiting from ART in the postpartum period and while breastfeeding.

21. In the Postpartum MTCT and Associated Factors section of the Results, the paragraph found in lines 233-236 states, “A total of 80.8% (21/26) of children with postpartum-MTCT had attended at least one unscheduled outpatient visits before the survey and a total of 26.9% (7/26) had been hospitalized, compared with the 55.0% (457/831) and 5.8% (48/831) of the HIV exposed uninfected children p-value=0.018 and p-value=0.001, respectively”, which suggests there may have been opportunities for postpartum infant/mother HIV diagnosis during this encounters. Do we know whether HIV-testing/diagnosis was done during these encounters?

Figure 1:

22. This entire figure could use some formatting/attention to detail. For example, some boxes have text aligned left while others are centered, some statements are incomplete, and some arrows are not well aligned.

23. The first exclusion (dashed arrow pointing to the right, just below the box that states “5000 randomly selected”) states “174 children >54 months at the”. This statement is incomplete, and it seems to be in conflict with what was reported in the text of the Results section (48 vs 54 months?).

24. The second exclusion (dashed arrow pointing to the left, just below the box that states “4826 children eligible”) states “17 houses not”. This statement is incomplete.

25. It’s still not clear to me why 350 (10%) of women and 61 (6.3%) of HIV-exposed infants have unknown HIV serostatus, when as HIV testing should have been performed per study protocol. Did these refuse testing? If so, this should be stated. If not, please provide some other explanation.

Table 1:

26. In the PDF, this table gets cut-off then continues on a second page. Might be easier to read/critique if re-formatted (landscape orientation, smaller font size, etc.).

27. At the top of the table, “Puesto como footnote” needs to be translated to English.

Table 2:

28. In the PDF, this table gets cut-off then continues onto three pages. Might be easier to read/critique if re-formatted (landscape orientation, smaller font size, etc.).

29. The p-value in the second row (“Mother located during the household survey”) is “1,000”. I’m assuming this is an error.

30. What is the source of the variables included in Table 2? This is not very well described in the Methods. In particular, was infant ARV prophylaxis based on self-report vs. medical records?

Table 3:

31. Again, in the PDF, this table gets cut-off then continues onto three pages. Might be easier to read/critique if re-formatted (landscape orientation, smaller font size, etc.).

32. Why weren’t “mother HIV diagnosis” and “mother ART initiation” included in the multivariable model. These were both strongly associated with postpartum MTCT in the univariate model. And, in terms of a priori assumptions, I would expect these to account for most of the MTCT risk. If the issue is missingness, then consider multiple imputation. This seems like a major flaw that needs to be addressed.

33. On the flip side of Comment #32, why was breastfeeding duration “forced” into multivariable model. It’s no surprise that breastfeeding duration wasn’t associated with MTCT, when the more important variables are timing of maternal HIV diagnosis and ART initiation; quantity is less important than quality (i.e., breastfeeding duration only becomes relevant when contextualized with timing of HIV diagnosis and ART initiation).

34. “Survey” is misspelled in “Age at survAy in months”.

Discussion:

35. The statement in the 4th paragraph of the Discussion (lines 290-292), “Our results suggest that in the context of B+ (lifelong ART to the mother and antiretroviral prophylaxis to the children), duration of breastfeeding is not associated with an increased risk of postpartum MTCT” is not entirely correct. It may in fact be true that duration of breastfeeding is not associated with an increased risk of postpartum MTCT when B+ guidelines are followed; but in this study the MTCT events seem to have occurred predominantly among women newly diagnosed with HIV and started on ART in the postpartum period. If you want to answer the question of impact of duration of breastfeeding in the context of B+, then you first have to control/adjust for timing of maternal HIV diagnosis and ART initiation (and ideally virologic control), both of which were strongly associated with MTCT in your univariable analysis but were not included in your multivariable analysis.

36. Likewise, the statement at the end of the 4th paragraph of the Discussion (lines 299-302) “Our findings reaffirm the 2016 guidelines on HIV-exposed infant feeding [14], which recommend that HIV-positive mothers who are well-controlled on ART should breastfeed their children for two years or more, as should HIV-negative mothers[14]” is an overstatement. Rather, than supporting WHO guidelines about prolonged breastfeeding (the veracity of which are not in question), this study instead highlights other very important issues, namely the need for ongoing HIV prevention, diagnosis, and ART initiation among women who remain at risk for HIV acquisition in the postpartum setting. I see this study as more of a justification for postpartum PREP and serially repeated HIV testing in postpartum period, rather than supporting prolonged breastfeeding. The authors start to touch on this in the 5th paragraph of the Discussion. All said, it seems as though the authors went into this analysis with a presupposition to prove (i.e., that prolonged breastfeeding is safe), rather than lettering the data speak for themselves. I strongly recommend reassessing the interpretation of the results and the messaging of the discussion and conclusions.

37. I have deferred further critiques of the discussion, until the Methodological issues and interpretation of the Results are addressed, as above.

Reviewer #2: Prolonged breastfeeding is safe in HIV-exposed children: Survey results in a high HIV prevalence community in southern Mozambique after implementation of Option B+

Fernandez-Luis et al.

This study looks at both duration of breastfeeding among HIV-exposed and HIV-unexposed infants as well as factors associated with postpartum transmission of HIV in Mozambique. HIV-exposed infants received six weeks of nevirapine, whether they were breastfeeding or not.

The authors conclude that women living with HIV breastfeed for a shorter median duration than women not living with HIV and that infants not receiving 6 weeks of nevirapine prophylaxis are at higher risk of HIV acquisition.

1) The study design is a bit confusing: women were chosen randomly to be interviewed but this was primarily a retrospective chart review combined with mother’s memories of what she did during pregnancy and postpartum.

2) What information do the authors have about maternal adherence to ART? In the era of U=U (undetectable equals untransmissible) in the world of sexual transmission, many obstetricians and pediatricians seek information on the effect of maternal adherence on transmission. Flynn et al have documented a 0.3% transmission rate at 6 months and 0.6% transmission rate at 12 months among women with undetectable viral loads. The Mozambique study addresses the “real world” of not having easy availability of viral loads but there is little said about maternal adherence. The authors cite a study by Pfeiffer 2017 that documented only 38% of women living with HIV still obtaining ART refills for themselves 90 days postpartum but do not comment on adherence among their study participants. Did their survey ask about adherence?

3) The actual risk of transmission appears to be 51% of 5.3%, or 2.7% (per Figure 1). That rate is significantly higher than the transmission rates in the PROMISE study (Flynn et al). Please comment.

4) I would also like some discussion of why the authors think that women living with HIV breastfed for a shorter period of time than women who did not have HIV. I can imagine that those with HIV may have been aware of some risk of transmission of HIV via breastmilk and, therefore, weaned their babies earlier. Did the surveys ask women why they stopped breastfeeding when they did? If not, what reasons do the authors think would explain early cessation?

5) If the standard of care in Mozambique is to give six weeks of infant prophylaxis, what were the reasons mothers gave for not giving it?

6) Of special interest to us in the U.S., Canada, and Europe, adding infant prophylaxis in this study was associated with a significant decrease in postpartum transmission. In high resource countries some pediatricians have opted to give nevirapine through cessation of breastfeeding in addition to continuing maternal ART. This study lends credence to that approach.

Although this study attempts to add important information to the literature, I am not sure that the study design allows us to confidently draw conclusions from it.

Reviewer #3: PLOS ONE

Please make objective more direct. "We aimed to compare" can be changed to "We compared...."

It is not clear what is meant by "Among the 5000 mother-child pairs selected, 69.7% (3486/5000) were found and

interviewed. " Please revise the terms "found" and "interviewed" to something like located, enrolled and surveyed.

Who were the initial 5000? All mothers who gave birth in the study interval? Please clarify.

The method used to estimate the mode of transmission should be specified in the methods of the abstract.

Line 16 - take out "In addition."

line 26 - updating does not need a hyphen

Line 56- change to just "We compared the duration..."

The introduction is excellently written and has great content.

Please explain how breastfeeding duration was obtained. Expand this section at is is the main dependent variable and currently not well explained in terms of how BF status and duration was obtained accurately. Reference a method please.

Mothers were surveyed, not interviewed. Or call this a structured interview. Survey is more appropriate since no qualitative data was obtained.

Results: do not have a sub-heading for sociodemographic characteristics. Make this frist first paragraph with no boldface.

Combine limitations in one paragraph.

Should the title be revised to include the dual object of the study which is to compare BF duration and MTCT transmission by ART prophylaxes status?

The conclusion is well written and supports the results. Well done!

6. PLOS authors have the option to publish the peer review history of their article (what does this mean?). If published, this will include your full peer review and any attached files.

Reviewer #1: No

Reviewer #2: No

Reviewer #3: No

---

## [Author Response · Author response to Decision Letter 0]

17 Mar 2022

Reviewer #1: General:

1. This study has the potential to make important contributions to the literature on this subject. With the success of PMTCT programs, vertical transmission rates have been declining; however, MTCT has not been eliminated, and the relative contribution of postpartum/breastfeeding MTCT to the vertical transmission events that continue to occur is not well understood. This study suggests that postpartum/breastfeeding MTCT most commonly occurs in the context of women newly acquiring HIV in the postpartum period and in the absence of infant ARV prophylaxis. The authors emphasize that duration of breastfeeding was not associated with postpartum MTCT, suggesting that prolonged breastfeeding can/should be supported in this context; however, this conclusion does not seem to logically follow from the results. Rather, it seems that the emphasis should be on prevention, diagnosis, and treatment of women who remain at risk for HIV acquisition in the postpartum period. It may be true that prolonged breastfeeding might be safe (assuming that the mother and baby are adherent to ART and ARV prophylaxis, respectively), but these data/findings do not strongly support this conclusion. That all said, the interpretation and messaging need to be modified. If these issues and the following comments are adequately addressed, then I think this article can reconsidered for publication.

Answer: Thank you for general comments. It is true that the natural conclusion that logically flows from the results is on prevention, diagnosis and treatment of women who remain at-risk for HIV acquisition. Thus, we modified the interpretation and messaging putting more emphasis on prevention, diagnosis, and treatment of women who remain at risk for HIV acquisition in the postpartum period and on infant ARV prophylaxis. In addition, we have softened the messaging on supporting prolonged breastfeeding for those mothers who started breastfeeding in LMIC situations where mother and baby are adherent to ART and ARV prophylaxis, respectively.

Title:

2. I appreciate the sentiment, but to declare prolonged breastfeeding “safe” without further context is too strong. Furthermore, the context was a cross-sectional sample of mother-child pairs, of whom nearly 30% could not be found and therefore had unknown (and perhaps worse) outcomes. Also, among the subset of HIV-exposed infants who were located and tested, there was an overall MTCT rate of 5.3%, more than half of which was attributed to postpartum/breastfeeding. All said, a postpartum/breastfeeding MTCT rate of 2.7% (26/967) should not be celebrated as “safe”. Just because self-reported breastfeeding duration was not associated with post-partum MTCT in your multivariable model, it does not mean that the postpartum MTCT was not attributable to breastfeeding; in fact, it seems that breastfeeding in the absence of maternal HIV diagnosis, maternal ART, and infant ARV prophylaxis might have been the cause for these MTCT events. Consider tempering the title and revising the interpretation and messaging throughout the manuscript.

Answer: Thank you for the comment, the title has been modified eliminating the word “safe” and reinforcing the importance of MTCT prevention. We have also revised the messaging throughout the manuscript.

Abstract:

3. The study design should be more clearly stated. From what I can gather, this is a cross-sectional assessment of an observational cohort.

Answer: Thank you for your comment. We have specified that it is a cross sectional assessment. 

4. 69.7% (3486/5000) were “found and interviewed”. Were all of these women and their children also tested for HIV, as suggested in the Methods. If so, what was the median age [IQR] of children when they were tested, and were all tested after complete cessation of breastfeeding?

Answer: 

Thank you for your comment. 

Mothers who do not know their status or self-report being HIV-negative were tested at survey, as well as the HIV-exposed children. For those children whose mother was not available and the main caregiver provided consent, age-appropriate testing was also offered. Documented known HIV-positive individuals were not re-tested, however a Geenius HIV-1/2 Confirmatory Assay was performed. This information has been added in the methods of the manuscript.

 Median age [IQR] of children at survey was 23.9 (14.9 - 35.7) months as described in table 1. 

5. How can 10% of children have unknown HIV exposure status when/if all of the mothers were tested for HIV in the context of this study (see Comment #4, above)? Perhaps this is made clear later in the manuscript.

Answer: We didn’t add this information in the abstract due to the limitations in the number of words. However, after your comment more information about these 350 mothers with unknown serostatus has been added in figure 1:

317 migration/absence (9.1%)

1 indeterminate (0.03%)

1 death (0.03%)

31 refusals (0.9%)

We have also included more information about the 61 children with Unknown HIV serostatus in figure 1:

15 migration/absence (1.5%)

4 death (0.4%)

21 refusals (2.2%)

21 no information (2.2%)

6. “P-value” does not need to be spelled out (e.g., p<0.001 should suffice).

Answer: thank you for your comment. We have changed P-value with p.

7. For the same reasons outlined in Comment #2 (re: Title), I do not think the Conclusion is well supported by the Results. To me, the results suggest that more needs to be done to promote uptake of infant ARV prophylaxis, and that perhaps when mothers are virally suppressed AND infants are appropriately prophylaxed, then prolonged breastfeeding can and should be supported in this context.

Answer: Thank you for your comment. As explained in previous responses, the conclusion has been revised. 

Introduction:

8. In the last sentence of the 5th paragraph (line 52) of the Introduction, there is a typo. Presumably, “2014-1015” should be “2014-2015”.

Answer: Thank you for your comment. It has been corrected. 

9. In the 5th paragraph of the Introduction, the authors state that “there is no data about ART adherence and viral load during breastfeeding period at national level…”, so they instead cite Reference #18 to illustrate that there is substantial non-adherence to ART refills in the first 3 months postpartum. Firstly, it should be “there ARE no data…” (plural). Secondly, aren’t there PEPFAR data about retention and viral suppression that can be disaggregated for pregnancy/breastfeeding status (or at least for women of child bearing age) to approximate national statistics?

Answer: typo has been corrected. We have added PEPFAR data about viral suppression in pregnancy. 

10. There is sufficient introduction to breastfeeding guidelines for HIV+ women, but there is insufficient background information about HIV-exposed infant ARV prophylaxis guidelines, which have also evolved over time. Since your main finding was that lack of infant ARV prophylaxis was associated with postpartum MTCT, it seems like more background on this needs to be presented (i.e., WHO and Mozambican guidelines).

Answer: thank you for the recommendation. We have added more background on ARV prophylaxis guidelines

Methods:

11. In the last paragraph of the Statistical Analysis section of the Methods (line 157), the authors state that there were “61 children with unknown HIV serostatus”, but in the Abstract it is noted that “for 10.1% (350/3486) HIV-exposure was unknown.” This distinction becomes clear once the reader gets to Figure 1, but please clarify this in the text as well.

Answer: Thank you for your comment. 

 There are 350 children with HIV-exposure unknown due to the status of the mother was not possible to determine. These 350 children who we didn’t know if were or not exposed to HIV were excluded of the analysis. In addition, among the HIV-exposed children, we have 61 children HIV-exposed with unknown HIV serostatus as it was not possible to perform HIV test in these children. 

We have specified that the 350 children with unknown HIV-exposures were excluded and we have also reworded the sentence about the 61 children with unknown HIV serostatus for more clarification. In addition, in figure 1, we have added information about the reasons for unknown HIV exposure and HIV status, respectively. 

Results:

12. In the Study Design and Study Population section of the Methods (line 84) it is stated that “5000 children born alive in the previous 48 months were randomly selected”, but in the first sentence of the Results section (line 174) it is stated that 174 children were ineligible/excluded because they were “older than 48 months”. How is this possible?

Answer: Thank you for your comment. It was mainly to an error in the date of birth in the HDSS database. During the informed consent process, the field workers of the study verified the inclusion criteria including the date of birth and found that those 174 children were not eligible to participate in the study. 

13. In the second sentence of the Results it is noted that “1340 [27%] mother/child pairs were not located”, but no further explanation is provided in the text. In Figure 1, it is noted that these were mostly “absent/migrant”, but this is not well characterized. Also in Table 1, among those who were included/located, it seems that all of those women who were “absent or migrated” were HIV+ (n=85; 8.8%) vs. HIV- (n=0), but the numbers are different for the HIV-exposed (n=15; 1.6%) vs unexposed infants (n=23; 1.1%). Whatever the case, this might suggest that these were not missing at random. Can the authors provide some additional details as to why mother-child pairs might not have been located, and whether the characteristics of those not located were similar/different to those who ultimately participated in the study? Is it possible that those who were not located were also more likely to be non-adherent to PMTCT care and therefore more likely to have postpartum MTCT? All said, I am concerned that the study findings might not be generalizable based on this limitation. This must be addressed or at least acknowledged as a severe limitation.

Answer: Thank you for the comment. We have added more information on the reasons that 27% of the mother/child pairs initially randomly selected were not located. We don’t have information about HIV serostatus of these mothers and we have added this as a limitation of the study. 

Please note that Table 1 is not related with these 27% mother/child pairs not located. Table 1 includes children born from mothers with known HIV status (967 HIV positive and 2169 negative). This table describes whether the mother was located and surveyed or not (and this means that the survey was performed with the caregiver). 

14. Very minor point, but in the Results (line 178) it is stated that for 10.0% HIV-exposure was unknown, while in the Abstract it says 10.1%. Please reconcile.

Answer: Thank you for your comment. It has been corrected

15. In line 181 it is stated that “13.4% (130/967) were exposed only in the postpartum period”. What proportion of these were characterized as such based on maternal HIV testing performed during the study visit? What proportion of these women were diagnosed with HIV and started on ART while the baby was still breastfeeding?

Answer: 

Thank you for the comment. Among the children exposed only in the postpartum period 47.6% were characterized as HIV-exposed based on maternal HIV testing performed during the study visit and 14% were still breastfeeding when their mothers started on ART. We have added this information. 

16. In lines 186-188 it is stated “HIV-positive mothers were significantly older, with a median age of 28.7 years (IQR: 23.4-187 33.4) and almost all (98.1%) had attended at least one antenatal visit, when compared with HIV-negative mothers.” Can you please include the statistics for the HIV-negative women for comparison, rather than requiring the reader to cross-reference Table 1.

Answer: Thank you for the suggestion. The statistics for HIV-negative women have been added. 

17. In line 189, it is stated that, “Only 34.1% of HIV-positive mothers had at least one viral load result at the time of survey.” What proportion of these were virologically suppressed?

Answer: Thank you for your question. Only 34.1%% (330/967) of HIV-positive mothers had at least one viral load result at the time of survey. However, 75.8% (250/330) of the mothers with viral load result were virally suppressed. We have added this information. 

18. In the Postpartum MTCT and Associated Factors section of the Results (line 208), “6,2%” should be “6.2%”.

Answer: Thank you for your comment. It has been corrected

19. In the Postpartum MTCT and Associated Factors section of the Results, the sentence “Among the 61 children with unknown HIV serostatus, 62.3% (38/61) were female, 4 never breastfed and among the 57 who initiated breastfeeding, the median duration of breastfeeding was 12 (95%CI: 10.4 - 15.1) months and a total of 53 would meet the definition of postpartum MTCT” (lines 212-215) is quite confusing and needs to be re-written to enhance clarity.

Answer: thank you for the comment. It has been corrected. 

20. In the Postpartum MTCT and Associated Factors section of the Results, the paragraph found in lines 227-232 is essential to contextualizing the findings of this study. The fact that, “Mothers with postpartum MTCT were mostly diagnosed (57.7%) and initiated ART (50.0%) after the child was born, compared with the 12.3% and 11.4% of diagnosis and ART initiation after the birth among the mothers of HIV exposed uninfected children, p-value<0.001” indicates a problem that heretofore has not been very well discussed. This frames the problem not only as postpartum MTCT among mother-child pairs breastfeeding in the context of known HIV status during the antepartum/peripartum period, but rather postpartum HIV diagnosis and ART initiation (and lack thereof) among women who were previously HIV-negative or HIV-unknown status. Similarly, that “only 50.0% (13/26) of children with postpartum MTCT had received antiretroviral prophylaxis at any time after birth compared to 82.8% (688/831) of HIV exposed uninfected children, p-value<0.001” should not come as a surprise when nearly two-thirds of their mothers were not diagnosed with HIV until after the child was born. This further emphasizes that the issue has less to do with prescription and adherence to infant ARV prophylaxis and more to do with mothers not knowing their HIV status and/or not benefiting from ART in the postpartum period and while breastfeeding.

Answer: thank you for your comment. We agree with the reviewer and we have highlighted in the discussion that “Mothers may not have initiated ART because of their lack of awareness of their HIV status, death, not willing or because of service delivery shortfalls or stockouts.” We have included data on HIV incidence in breastfeeding women during the postpartum period in Mozambique and we have suggested that “Our results suggest that improving the reach of PrEP among women who remain at risk for HIV acquisition in the postpartum period, as well as establishing specific retesting times during the postpartum period in areas of high HIV incidence would reinforced the prevention of MTCT in LMIC and would facilitate initiation of ART in breastfeeding mothers and antiretroviral prophylaxis in their HIV exposed infants.” 

21. In the Postpartum MTCT and Associated Factors section of the Results, the paragraph found in lines 233-236 states, “A total of 80.8% (21/26) of children with postpartum-MTCT had attended at least one unscheduled outpatient visits before the survey and a total of 26.9% (7/26) had been hospitalized, compared with the 55.0% (457/831) and 5.8% (48/831) of the HIV exposed uninfected children p-value=0.018 and p-value=0.001, respectively”, which suggests there may have been opportunities for postpartum infant/mother HIV diagnosis during this encounters. Do we know whether HIV-testing/diagnosis was done during these encounters?

Answer: Thank you for your comment. We agree with the reviewer that that information would be very interesting, however this information is unfortunately unavailable. We have added in the discussion that these are potential opportunities for HIV testing/Diagnosis. 

Figure 1:

22. This entire figure could use some formatting/attention to detail. For example, some boxes have text aligned left while others are centered, some statements are incomplete, and some arrows are not well aligned.

Answer: Thank you for the observation. The figure has been corrected. 

23. The first exclusion (dashed arrow pointing to the right, just below the box that states “5000 randomly selected”) states “174 children >54 months at the”. This statement is incomplete, and it seems to be in conflict with what was reported in the text of the Results section (48 vs 54 months?).

Answer: Thank you for the observation. The figure and the text of the results section have been corrected. 

24. The second exclusion (dashed arrow pointing to the left, just below the box that states “4826 children eligible”) states “17 houses not”. This statement is incomplete.

Answer: Thank you for the observation. The figure has been corrected.

25. It’s still not clear to me why 350 (10%) of women and 61 (6.3%) of HIV-exposed infants have unknown HIV serostatus, when as HIV testing should have been performed per study protocol. Did these refuse testing? If so, this should be stated. If not, please provide some other explanation.

Answer: Thank you for the observation, the information has been added. 

Table 1:

26. In the PDF, this table gets cut-off then continues on a second page. Might be easier to read/critique if re-formatted (landscape orientation, smaller font size, etc.).

Answer: Thank you for your comment. It has been sent in excel format.

27. At the top of the table, “Puesto como footnote” needs to be translated to English.

Answer: Thank you for the observation. It has been removed. 

Table 2:

28. In the PDF, this table gets cut-off then continues onto three pages. Might be easier to read/critique if re-formatted (landscape orientation, smaller font size, etc.).

Answer: Thank you for your comment. It has been sent in excel format.

29. The p-value in the second row (“Mother located during the household survey”) is “1,000”. I’m assuming this is an error.

Answer: The p-value is 1.000, we have corrected it. 

30. What is the source of the variables included in Table 2? This is not very well described in the Methods. In particular, was infant ARV prophylaxis based on self-report vs. medical records?

Answer: thank you for the question. Sociodemographic (Age, Educational level, Marital status, Income, Parity), antenatal care and breastfeeding information were self-reported during the survey. HIV data were obtained through medical documentation (Mother HIV diagnosis, Mother ART initiation) or HIV database (Mother cd4 at child birth, Mother viral load at child birth). Hospitalizations and outpatients’ visits were obtained through HDSS database. Gestational Age and Infant Antiretroviral prophylaxis were obtained through medical documentation. This information has been added in methods section. 

Table 3:

31. Again, in the PDF, this table gets cut-off then continues onto three pages. Might be easier to read/critique if re-formatted (landscape orientation, smaller font size, etc.).

Answer: Thank you for your comment. It has been sent re-formatted. 

32. Why weren’t “mother HIV diagnosis” and “mother ART initiation” included in the multivariable model. These were both strongly associated with postpartum MTCT in the univariate model. And, in terms of a priori assumptions, I would expect these to account for most of the MTCT risk. If the issue is missingness, then consider multiple imputation. This seems like a major flaw that needs to be addressed.

Answer: The initial reason for not including “mother ART initiation” was more than 20% missing values and because “mother HIV diagnosis” was eliminated in the forward stepwise selection. However, after your recommendation we have performed the regression without using the stepwise selection process and we have performed multiple imputation on the “mother ART initiation” variable treated as a binary variable: ART initiation before delivery yes/no. We have assumed that missing data are missing at random and used a logistic regression model as a method to impute. In the imputation model we have included our outcome variable and the other predictor variables included in the final multivariable model and which have p>0.2 in the univariable model and have less than 20% missing value. We have made a total of 20 imputations. Using these parameters for the additional analysis, we obtained similar results with no observation of evidence of association between breastfeeding duration and postpartum MTCT. In contrast, mother ART initiation after the date of childbirth was associated with postpartum MTCT. We have not added “mother HIV diagnosis” in the regression because of collinearity with “mother ART initiation.” We feel this additional step to our analysis has made our results more robust and thank the reviewer for the suggestion. 

33. On the flip side of Comment #32, why was breastfeeding duration “forced” into multivariable model. It’s no surprise that breastfeeding duration wasn’t associated with MTCT, when the more important variables are timing of maternal HIV diagnosis and ART initiation; quantity is less important than quality (i.e., breastfeeding duration only becomes relevant when contextualized with timing of HIV diagnosis and ART initiation).

Answer: breastfeeding was forced because is the variables that we are specifically testing as part of this study.

34. “Survey” is misspelled in “Age at survAy in months”.

Answer: Thank you for your comment. It has been corrected

Discussion:

35. The statement in the 4th paragraph of the Discussion (lines 290-292), “Our results suggest that in the context of B+ (lifelong ART to the mother and antiretroviral prophylaxis to the children), duration of breastfeeding is not associated with an increased risk of postpartum MTCT” is not entirely correct. It may in fact be true that duration of breastfeeding is not associated with an increased risk of postpartum MTCT when B+ guidelines are followed; but in this study the MTCT events seem to have occurred predominantly among women newly diagnosed with HIV and started on ART in the postpartum period. If you want to answer the question of impact of duration of breastfeeding in the context of B+, then you first have to control/adjust for timing of maternal HIV diagnosis and ART initiation (and ideally virologic control), both of which were strongly associated with MTCT in your univariable analysis but were not included in your multivariable analysis.

Answer: Thank you for your suggestion. We have adjusted for maternal HIV diagnosis and ART initiation using multiple imputation in the regression analysis of postpartum MTCT. Due to >60% of women lacking a viral load result, we were unable to impute and adjust for viral load.

36. Likewise, the statement at the end of the 4th paragraph of the Discussion (lines 299-302) “Our findings reaffirm the 2016 guidelines on HIV-exposed infant feeding [14], which recommend that HIV-positive mothers who are well-controlled on ART should breastfeed their children for two years or more, as should HIV-negative mothers[14]” is an overstatement. Rather, than supporting WHO guidelines about prolonged breastfeeding (the veracity of which are not in question), this study instead highlights other very important issues, namely the need for ongoing HIV prevention, diagnosis, and ART initiation among women who remain at risk for HIV acquisition in the postpartum setting. I see this study as more of a justification for postpartum PREP and serially repeated HIV testing in postpartum period, rather than supporting prolonged breastfeeding. The authors start to touch on this in the 5th paragraph of the Discussion. All said, it seems as though the authors went into this analysis with a presupposition to prove (i.e., that prolonged breastfeeding is safe), rather than lettering the data speak for themselves. I strongly recommend reassessing the interpretation of the results and the messaging of the discussion and conclusions.

Answer: Thank you for your comment. We have reoriented the discussion and conclusions to highlight our results in the context of the need for ongoing HIV prevention, diagnosis, and ART initiation in the postpartum setting, including postpartum PREP. 

37. I have deferred further critiques of the discussion, until the Methodological issues and interpretation of the Results are addressed, as above.

Answer: Thank you for the comment. We have addressed all the comments above. 

 

Reviewer #2: Prolonged breastfeeding is safe in HIV-exposed children: Survey results in a high HIV prevalence community in southern Mozambique after implementation of Option B+

Fernandez-Luis et al.

This study looks at both duration of breastfeeding among HIV-exposed and HIV-unexposed infants as well as factors associated with postpartum transmission of HIV in Mozambique. HIV-exposed infants received six weeks of nevirapine, whether they were breastfeeding or not. 

The authors conclude that women living with HIV breastfeed for a shorter median duration than women not living with HIV and that infants not receiving 6 weeks of nevirapine prophylaxis are at higher risk of HIV acquisition.

1) The study design is a bit confusing: women were chosen randomly to be interviewed but this was primarily a retrospective chart review combined with mother’s memories of what she did during pregnancy and postpartum.

Answer: thank you for tour comment. We have reworded methods section in order to facilitate the compression. 

 It this cross-sectional household survey conducted from October-2017 to April-2018. Mothers who had given birth within the previous 48-months in the Manhiça district were randomly selected to be surveyed and to receive an HIV-test along with their children. During the survey, study HIV counselors administered a specific questionnaire designed to capture sociodemographic characteristics, HIV testing history and ART, antenatal care and duration of breastfeeding. Clinical documentation was used to obtain information about HIV testing and HIV care during pregnancy and postpartum. The mother or caregiver self-reported the total duration of any breastfeeding in months at the time of survey. 

2) What information do the authors have about maternal adherence to ART? In the era of U=U (undetectable equals untransmissible) in the world of sexual transmission, many obstetricians and pediatricians seek information on the effect of maternal adherence on transmission. Flynn et al have documented a 0.3% transmission rate at 6 months and 0.6% transmission rate at 12 months among women with undetectable viral loads. The Mozambique study addresses the “real world” of not having easy availability of viral loads but there is little said about maternal adherence. The authors cite a study by Pfeiffer 2017 that documented only 38% of women living with HIV still obtaining ART refills for themselves 90 days postpartum but do not comment on adherence among their study participants. Did their survey ask about adherence?

Answer: Unfortunately, adherence was not asked during the survey. We have highlighted more the importance of adherence in the discussion, according to your suggestion. 

3) The actual risk of transmission appears to be 51% of 5.3%, or 2.7% (per Figure 1). That rate is significantly higher than the transmission rates in the PROMISE study (Flynn et al). Please comment.

Answer: thank you for your suggestion. We have added the comment in the discussion. 

4) I would also like some discussion of why the authors think that women living with HIV breastfed for a shorter period of time than women who did not have HIV. I can imagine that those with HIV may have been aware of some risk of transmission of HIV via breastmilk and, therefore, weaned their babies earlier. Did the surveys ask women why they stopped breastfeeding when they did? If not, what reasons do the authors think would explain early cessation?

Answer: Thank you for your comment. The reason for stopped breastfeeding was not asked in the survey. We have added some reasons described in the literature as advice of health workers, influence of relatives, stigma, conflicting opinions about the risk for MTCT and poor dissemination of policies.

5) If the standard of care in Mozambique is to give six weeks of infant prophylaxis, what were the reasons mothers gave for not giving it?

Answer: Thank you for your comment. As explained in the discussion, we hypothesized that mothers may not have administered prophylaxis to their infants because of non-adherence, because of their lack of awareness of their HIV status or because of service delivery shortfalls or stockouts. We supported our hypothesis with a reference who showed high incidence of HIV in breastfeeding women during the postpartum period in Mozambique. 

6) Of special interest to us in the U.S., Canada, and Europe, adding infant prophylaxis in this study was associated with a significant decrease in postpartum transmission. In high resource countries some pediatricians have opted to give nevirapine through cessation of breastfeeding in addition to continuing maternal ART. This study lends credence to that approach.

Answer: thank you for your suggestion. We agree. However, other reviewer asks for repeat the regression after multiple imputation of the variable mother ART initiation. After the inclusion of mother ART initiation in the regression, we have not found association between infant prophylaxis and MTCT and for this reason we have not added the information about infant prophylaxis in high income countries. 

Although this study attempts to add important information to the literature, I am not sure that the study design allows us to confidently draw conclusions from it.

Answer: Thank you for the comment. We have reworded the conclusion.

 

Reviewer #3: PLOS ONE

Please make objective more direct. "We aimed to compare" can be changed to "We compared...."

Answer: thank you for your suggestion. It has been changed. 

It is not clear what is meant by "Among the 5000 mother-child pairs selected, 69.7% (3486/5000) were found and interviewed.

 " Please revise the terms "found" and "interviewed" to something like located, enrolled and surveyed.

Answer: thank you for your suggestion. We have used the words “located” and “enrolled” according to your suggestion. 

Who were the initial 5000? All mothers who gave birth in the study interval? Please clarify.

Answer: Thank you for your comment. The initial 5000 mother-children pairs were randomly selected among all children born alive in the previous 48 months at the HDSS. The sentence has been reworded to clarify. 

The method used to estimate the mode of transmission should be specified in the methods of the abstract.

Answer: Thank you for your suggestion. It has been added. 

Line 16 - take out "In addition."

Answer: Thank you for your suggestion. It has been removed. 

line 26 - updating does not need a hyphen

Answer: Thank you for your suggestion. It has been removed. 

Line 56- change to just "We compared the duration..."

Answer: thank you for your suggestion. It has been changed.

The introduction is excellently written and has great content.

Answer: Thank you for your comment. 

Please explain how breastfeeding duration was obtained. Expand this section at is is the main dependent variable and currently not well explained in terms of how BF status and duration was obtained accurately. Reference a method please.

Answer: thank you for your comment. The mother or caregiver self-reported the total duration of any breastfeeding at the time of survey. This information has been included in methods. 

Mothers were surveyed, not interviewed. Or call this a structured interview. Survey is more appropriate since no qualitative data was obtained.

Answer: thank you for your comment. It has been corrected. 

Results: do not have a sub-heading for sociodemographic characteristics. Make this frist first paragraph with no boldface.

Answer: sub-heading for sociodemographic characteristics has been deleted according your recommendation. 

Combine limitations in one paragraph.

Answer: limitations have been combined in one paragraph, according to your suggestion. 

Should the title be revised to include the dual object of the study which is to compare BF duration and MTCT transmission by ART prophylaxes status?

Answer: thank you for the comment, title has been modified. 

The conclusion is well written and supports the results. Well done!

Answer: thank you for the comment.

---

## [Editor Report · Decision Letter 1]

30 May 2022

Prompt HIV diagnosis and treatment in postpartum women is crucial for prevention of mother to child transmission during breastfeeding: Survey results in a high HIV prevalence community in southern Mozambique after the implementation of Option B+.

PONE-D-21-25758R1

Dear Dr. Fernandez,

We’re pleased to inform you that your manuscript has been judged scientifically suitable for publication and will be formally accepted for publication once it meets all outstanding technical requirements.

Kind regards,

Rena C Patel, MD, MPH

Academic Editor

PLOS ONE

Additional Editor Comments (optional):

Dear authors, I greatly appreciate that you have taken reviewer comments to heart in revising this manuscript, with significant changes in the methods (e.g., MI, modeling, etc.) and in reporting (write-up of methods and discussion). I still note some minor copy editing issues that can, hopefully, be corrected at that stage. I would like to personally apologize for the delay in reaching a decision of the work. Thank you, Rena
---

## [Editor Report · Acceptance letter]

20 Jul 2022

PONE-D-21-25758R1 

Prompt HIV diagnosis and treatment in postpartum women is crucial for prevention of mother to child transmission during breastfeeding: Survey results in a high HIV prevalence community in southern Mozambique after the implementation of Option B+. 

Dear Dr. Fernandez:

I'm pleased to inform you that your manuscript has been deemed suitable for publication in PLOS ONE. Congratulations! Your manuscript is now with our production department. 

Kind regards, 

on behalf of

Dr. Rena C Patel 

Academic Editor

PLOS ONE